# Infrared spectroscopy data- and physics-driven machine learning for characterizing surface microstructure of complex materials

Joshua L. Lansford [1] & Dionisios G. Vlachos [1,2]

There is a need to characterize complex materials and their dynamics under reaction conditions to accelerate materials design. Adsorbate vibrational excitations are selective to adsorbate/surface interactions and infrared (IR) spectra associated with activating adsorbate vibrational modes are accurate, capture details of most modes, and can be obtained operando. Current interpretation depends on heuristic peak assignments for simple spectra, precluding the possibility of obtaining detailed structural information. Here, we combine data-based approaches with chemistry-dependent problem formulation to develop physics-driven surrogate models that generate synthetic IR spectra from first-principles calculations. Using synthetic IR spectra of carbon monoxide on platinum, we implement multinomial regression via neural network ensembles to learn probability distributions functions (pdfs) that describe adsorption sites and quantify uncertainty. We use these pdfs to infer detailed surface microstructure from experimental spectra and extend this methodology to other systems as a first step towards characterizing complex interfaces and closing the materials gap.

[1] Department of Chemical Biomolecular Engineering, University of Delaware, 150 Academy Street, Newark, DE 19716, USA. [2] Catalysis Center for Energy Innovation, University of Delaware, 221 Academy Street, Newark, DE 19716, USA. ✉email: vlachos@udel.edu

First-principles computations are making major inroads in advancing structure-property relations and providing insights into in silico prediction of materials. This model-driven approach has, however, had limited success for heterogeneous materials consisting of multiple surface sites, such as nanoparticles of various sizes and shapes, and/or of spatially inhomogeneously distributed elements, such as bimetallics and high entropy alloys. Solid catalysts used for electrochemical and thermochemical transformations, ranging from fuel cells to production of fuels and chemicals, are an important class of these materials. Compounding this difficulty in understanding, materials change dynamically in response to their environment, and thus, their ex situ characterization is often of limited value. The total number of surface sites of many metals is often quantified by CO chemisorption. Calorimetry and temperature programmed desorption (TPD) provide indirect and rather coarse-grained surface characterization[1,2]. Thus, the type of sites, as well their prevalence and contribution to catalyst performance remain largely unknown. First-principles models most often consider only one active site on a crystallographic plane, closely mimicking ultra-high vacuum single crystal experiments. The disparity between single crystal experiments, and associated calculations, and real-world materials is known as the materials gap[3–5]. Our ability to close the materials gap demands methods to quantify the types of surface sites of complex materials, along with their dynamic behavior under operando (working) conditions. This is an active research direction of both government funding agencies and private companies[6,7].

Toward characterizing the structure of real nanomaterials, reverse Monte Carlo analysis of X-ray absorption fine structure spectroscopy (EXAFS) data has proven successful in resolving the structure of bimetallic nanoparticles with atomic resolution[8]. Neural networks, trained on X-ray absorption near edge structure (XANES) data, predict average coordination numbers of coordination shells[9] and radial distribution functions[10] given experimental spectra from monometallic nanoparticles. XAS spectroscopy is, though, a bulk technique[11,12], whereas many phenomena, such as catalysis, depend directly on surface properties. Adsorbate vibrational excitations are, on the other hand, selective to adsorbate/surface interactions[11]. Infrared (IR) spectroscopy is commonly employed for characterizing adsorbate/surface, gas, and liquid-phase vibrational transitions. Fourier Transform IR (FTIR) spectroscopy with femtosecond time resolution can infer the structure of electronically excited transition metal complexes[13], while two-dimensional spectroscopy tracks chemical transition states[14] and coupling between vibrational modes[15] in liquids. For solid surfaces, broad spectrum IR with nanometer spatial resolution is possible[16]. A major advantage of IR is that the spectra are very accurate[17], capture details of most vibrational modes[18,19] including coverage effects[2,20,21], and can be obtained quickly in situ or operando for many environments[22,23]. Most IR-based peak assignments are heuristic and can be applied only to relatively simple spectra. Recent advances in IR-based quantification involve site-specific extinction coefficients[24] in conjunction with peak deconvolution, integration, and a priori assumptions about particles sizes and adsorbate coverage distribution[23]. Due to the expense of first-principles calculations[25], their direct use for detailed site and coverage identification would require generation and computation on a combinatorial number of structures to match spectra; this random match is beyond current and future computational power.

Here, we introduce a first-principles quantitative surface-selective IR methodology and integrate it with data-based approaches, chemistry-dependent problem formulation, and experimental data toward closing the materials gap to predict surface sites with atomic resolution from experimental data.

Throughout the rest of this paper we refer to chemistry-dependent problem formulation and application of relevant physics as expert knowledge. We quantify error in both C–O and Pt–C frequencies of chemisorbed CO on Pt nanoparticles and extended surfaces and discover that density functional theory (DFT) generated spectra, even at high coverage, are much more accurate for determining adsorption site and deducing local microstructure than DFT energies. Our method untangles site-specific molecule/surface interactions, interprets complex experimental IR spectra, and quantitatively infers type and number of surface sites and adsorbate coverage. DFT-computed frequencies and intensities at low CO coverage serve as primary data; we feed this data through layers of physics-driven surrogate models to generate a secondary dataset of synthetic IR spectra to describe an arbitrary combination of adsorption sites and ultimately deduce structure. For each set of DFT frequencies and intensities, we quantify adsorption sites using both the binding-type (atop, bridge, threefold or fourfold) and the generalized coordination number (GCN)[26]; microstructure is then described using binding-type and GCN probability distribution functions (pdfs) from experimental IR data. We derive a closed-form solution for the derivative of the squared Wasserstein distance with respect to the softmax activation as a finite sum; this enables us to train two separate neural network ensembles to learn binding-type and GCN pdfs from synthetic spectra and quantify error in the predictions. Together, we refer to both neural network ensemble models as the structure surrogate model. We evaluate the structure surrogate model on both synthetic and experimental IR spectra, develop software, and implement the methodology with both CO and NO as probe molecules.

## Results

**Modeling overview.** As there are many distinct methods developed in this work, we provide an overview of how we generate complex synthetic spectra in this section. We reference each part of the spectral surrogate model and refer to the relevant sections of the Methods and Supporting Information (SI) Notes. First, a schematic of the materials gap and general methodology for addressing this gap are given in Fig. 1a, b, respectively. In Fig. 1b, physics-driven surrogate models generate frequencies and intensities from DFT, screen outliers, apply coverage scaling, and perform spectral convolution; data-driven models involve class/group labeling and neural network training.

Methods' Section 1 provides an explanation for CO as an ideal probe molecule for developing the first spectral-based model. DFT setup is provided in Methods Sections 2–4. Details of the binding-types, calculating surface GCN, and tabulating occupied sites corresponding to any individual complex spectra into binding-type and GCN pdfs are covered in Methods' Section 5 and SI Notes 3–4. Details for calculating frequencies and intensities for adsorbates at an individual site can be found in Methods' Section 6 with frequency scaling factors accounting for errors from the harmonic approximation explained in Methods' Section 7 and SI Note 1. To account for lateral interactions, Methods Section 8 and SI Note 2 outline coverage scaling factors (CSF) computed from extended surface calculations. The fundamental nature and universality of coverage scaling to both frequencies and intensities are discussed in the main text. To mimic all possible real spectra coming from adsorption at a variety of sites on different size and shaped nanoparticles, Methods' Section 9 and SI Note 4 illustrate how individual simple spectra are mixed in different proportions. Section 10 of the Methods outlines how complex spectra are convoluted with slightly different parameters. In SI Note 10, all parts of the

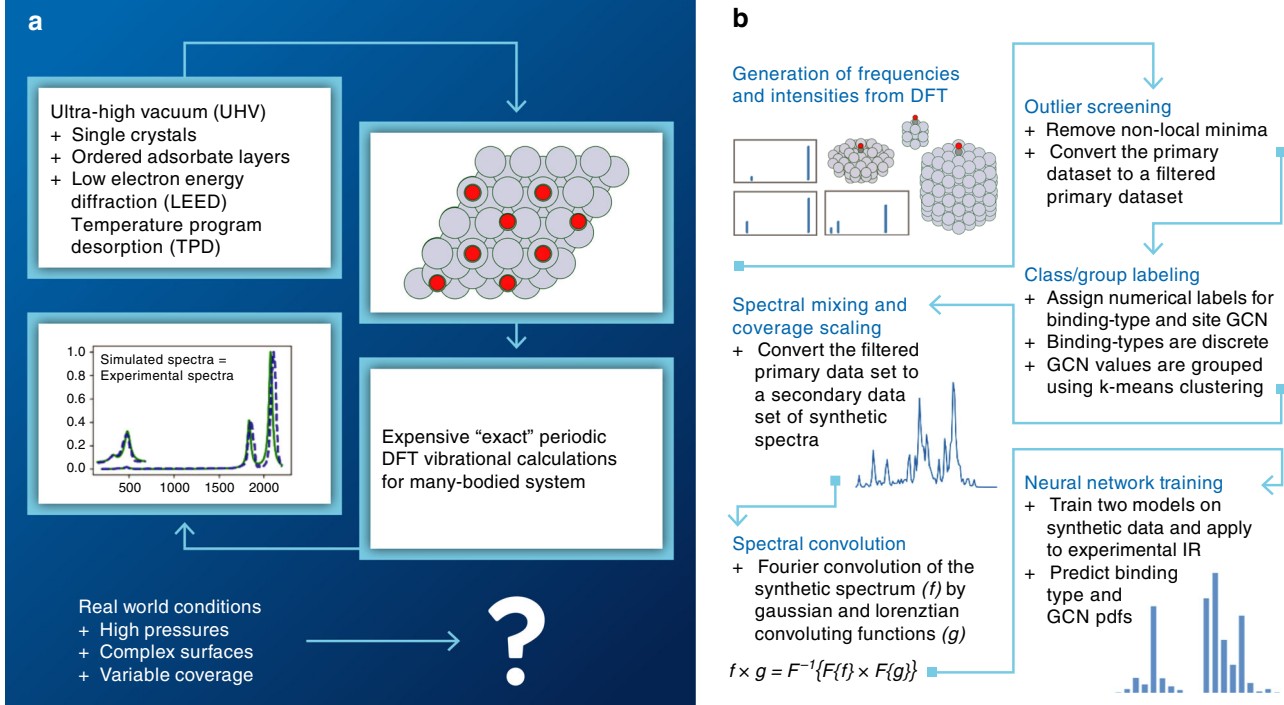

**Fig. 1 Approach combining expert knowledge and spectroscopic data to close the materials gap.** The materials gap (**a**) and corresponding workflow (**b**) to address this gap. **a** Surface structure with adsorbates can be characterized only in controlled and limited environments. Materials used in industrially relevant catalysis have no reliable characterization method to quantify the structure of the surface in detail. This gap between studies that can be performed with model catalysts and those on complex materials is known as the materials gap. **b** Develops a workflow to address this gap via physics- and data-driven surrogate models. DFT generates spectra of single CO molecules chemisorbed on different sites of many Pt nanoparticles. After non-local minima (outliers) are removed, a data-driven approach is used to assign labels to each data point that describe structure. The binding-type (atop, bridge, threefold, and fourfold) is naturally discrete and given a numerical class value 1–4. The coordination environment of the site is quantified by its GCN value, which is nearly continuous. K-means clustering is an unsupervised learning method that assigns GCN values to discrete GCN groups necessary for data-driven structure quantification. A physics-driven surrogate model is then applied to expand the filtered primary DFT-based dataset to a secondary dataset of complex spectra. This surrogate model includes coverage scaling factors that quantitatively link shifts in frequencies and intensities of the primary filtered data set to spatial coverage. The coverage scaling is done simultaneously with spectral mixing where individual simple-CO spectra are summed to enforce physics-driven coverage constraints. The last step in the physics-driven surrogate model is spectral convolution accomplished by a Fourier transform to generate synthetic complex spectra with varying line widths. Hundreds of thousands of complex spectra associated with various distributions of occupied sites, with varying coverage and differing line widths, are generated. In generating complex spectra, the binding-type and GCN group of the individual spectra are tallied and used to generate binding-type and GCN probability distribution functions (pdfs) that correspond to each complex spectrum. A data-driven surrogate model is trained on these synthetic complex spectra to learn microstructure. After training on synthetic complex spectra, the model is applied to experimental spectra.

approach to addressing the materials gap outlined in Fig. 1 are extended to systems with NO as a probe molecule.

**First-principles data and physical models.** To increase generalizability of our structure surrogate model, we select a diverse set of structures (Pt clusters and nanoparticles) and adsorption sites to expose multiple coordination environments representing facets, defects (adatoms and vacancies), edges, and corners. Subsequently, we perform DFT calculations of a single-adsorbed CO on sites of these structures. Forces and dipole moments are used to compute IR frequencies and intensities, respectively, from this DFT data under the harmonic approximation. Our CO primary DFT dataset consists of 1090 unique sets of frequencies and intensities. We have similar primary DFT datasets described in the SI Note 10 consisting of NO chemisorbed on Pt. The computations and the DFT setup are described in the Methods section.

Structural data imparted into spectra account for CO binding on different binding-type (atop, bridge, threefold, and fourfold sites) and coordination environments, e.g., different planes and edges. The former (binding-type) is defined by the number of Pt

atoms in the first coordination sphere of a chemisorbed CO molecule. The latter (site coordination) is described with the GCN[26]. The GCN is a weighted average that effectively describes the number of Pt atoms bonded to the adsorption site[27]. Figure 2 depicts IR frequencies and corresponding intensities for CO adsorbed at either atop or bridge sites of different GCN on similarly sized Pt nanoparticles. The data supports the well-known fact that spectroscopic signatures are sensitive to the coordination environment[28]. We find this to be true of both the Pt–CO and the C–O frequencies. The intensities vary much more with binding-type and GCN than the frequencies themselves.

In order to correct for systematic errors in calculated frequencies, scaling factors are commonly computed for gas-phase molecules[29]. Here we carry out the first benchmarking of adsorbate frequencies between first-principles calculations and experiments. In the Methods section, we formulate frequency scaling factors for chemisorbed CO on Pt that, when applied to DFT-calculated frequencies, result in mean absolute errors (MAE) in the C–O and Pt–CO stretch frequencies with respect to experiment of 3.87 and 1.49 cm$^{-1}$, respectively. These errors correspond to 0.19% and 0.33% of the average experimental C–O

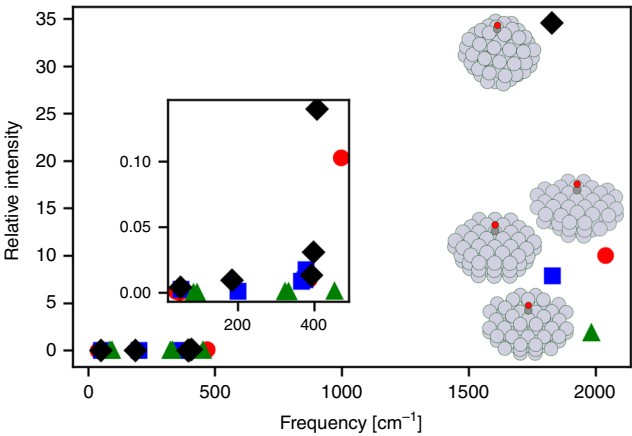

**Fig. 2 DFT frequencies and corresponding intensities for an adsorbed CO on different sites.** Data for atop site of $Pt_{68}$ with a GCN of 7.5 (red circles), atop site of $Pt_{78}$ with a GCN of 8.42 (green triangles), bridge site of $Pt_{72}$ with a GCN of 7.33 (blue squares), and bridge site of $Pt_{79}$ with a GCN of 3.5 (black diamonds). The subscript refers to the number of Pt atoms. The inset is a zoomed in image of the Pt–CO frequencies.

and Pt–CO frequencies in our dataset (2019 and $446\,cm^{-1}$), respectively. Although the average error is small, individual C–O frequency errors range from $-8.92$ to $11.87\,cm^{-1}$. However, it is possible that experimental error (including uncertainty in the actual coverage) could be a contributing factor to data points with large error. The dataset is located in Supplementary Tables 1 and 2. In addition to describing chemisorbed CO frequencies well, the relative intensities for chemisorbed CO spectra can be computed reliably from DFT despite significant errors in DFT-calculated CO adsorption energies[28,30,31]. CO frequencies do not correlate with adsorption energy as the governing physics for each is different (see Supplementary Fig. 14). DFT frequencies below $200\,cm^{-1}$ can have higher relative error due to breakdown of the harmonic approximation and coupling of the surface phonons. We find however, for the frustrated rotations of CO on Pt(111), the DFT-calculated frequencies (52 and $56\,cm^{-1}$) agree well with the value measured from experiment using He scattering ($48.5\,cm^{-1}$)[32]. Furthermore, these modes have low IR intensity and thus do not significantly affect the IR spectra of chemisorbed CO. Supplementary Fig. 1 demonstrates that the DFT-spectra for CO on Pt(111) in a $c(4 \times 2)$ configuration at 0.5 monolayers (ML) compares well with the experimental spectra. In both accuracy and resolution, spectroscopic signatures (DFT and/ or experimental IR) win over energy-based methods (DFT energies, chemisorption, TPD, and calorimetry) for closing the materials gap. A challenge DFT-based vibrational calculations face is that computational time becomes prohibitively high for very large molecules.

**Outliers and visualization of the primary dataset.** As a first step in data analytics, we employ visualization, a simple yet powerful way to evaluate correlations and identify outliers in data with a small number of descriptors. The primary dataset, visualized in Fig. 3a, includes CO at atop (green circles), bridge (red squares), threefold (yellow triangles), and fourfold (blue diamonds) adsorption sites. Plotting the GCN against the C–O frequency reveals many outliers. Statistical outliers inhibit learning via regression because they result in large gradients during training[33] and inadequate number of samples with similar features for prediction. While there are many methods to remove statistical outliers[34–36], these methods cannot distinguish between unphysical and physical outliers. Specifically, materials that are physical

outliers could possess superior performance and be the most interesting; such data should be kept and analyzed. Here, removing samples that are not local minima on the potential energy surface, such as transition states and saddle points (identified by at least one imaginary frequency), leverages expert knowledge to remove unphysical outliers.

After removing outliers from the primary dataset, the resulting filtered primary dataset consists of 878 local minima (Fig. 3b). Several interesting features are seen. First, the data is clustered primarily on binding-type, with the frequency increasing as the number of CO–Pt bonds decreases, from a low value for fourfold binding to a high value for atop binding, corresponding to the degree of orbital overlap between the CO $1\pi$ orbitals with the Pt d-band[28]. Second, and more interesting, sites with widely varying GCN can have similar C–O frequencies (clustered together). This clustering underlines the fact that the C–O frequency alone is inadequate to determine the microenvironment of a site. Since the C–O frequency is traditionally the only one used in IR characterization[11,37], this imposes a challenge in surface site identification and in closing the materials gap.

To overcome the limitations of C–O frequency as the sole descriptor of microstructure, we exploit additional spectroscopic signatures. Figure 3c, d shows the Pt–CO frequency and C–O intensity, respectively, against the C–O frequency with the GCN value indicated in the color bar. Improved separation of data with similar GCN values is visible. Figure 3c, d also reveals regions where increased sampling would improve model accuracy (a task pursued here only manually). For example, there are relatively few adsorption sites in our data possessing simultaneously low C–O frequencies and high GCN values.

**Coverage effects on spectroscopic signatures.** The CO coverage is often far from being low. Furthermore, CO may be distributed inhomogeneously by forming islands[2,38] and preferentially adsorbing at low GCN defect sites[39], even at low total coverage. Due to the computational expense of DFT, the filtered primary dataset was obtained in the low coverage regime, i.e., one CO molecule per cluster or nanoparticle. A method of extending low coverage spectra to high and variable coverage spectra is clearly needed. Experiments on CO on extended Pt surfaces[40,41], nano-particles[42], and electrodes[43] indicate that lateral interactions affect the C–O stretch frequency. These experimental studies have revealed that the C–O spectra peak-positions scale linearly with coverage; yet, rigorous quantification of coverage effects to frequencies is lacking and effects to intensities are almost completely ignored due to difficulty in experimentally quantifying intensity changes with coverage. For determining coverage distribution on sites of different binding-type and GCN, we show here that intensities are crucial.

DFT calculations at different total spatial coverages on 111, 100, and 110 low-index planes of CO at the atop (circles), bridge (squares), threefold (triangles) and fourfold (diamonds) sites reveal universal linear scaling of C–O and Pt–CO frequencies (Fig. 4 and Supplementary Fig. 2a) irrespective of the coordination of the site.

Spatial coverage of identical sites is achieved by emulating different binding sites using different isotopes with very different masses, explained in SI Note 2. Interestingly, the dependence of the IR data on coverage is a function of the binding-type only and independent of the GCN, possibly due to reduction of overlap between the CO $1\pi$ orbitals and metal d-band with increasing coverage that varies much more with binding-type than with GCN. Further work is needed to fully understand the origins of these coverage scaling relationships as well as why the C–O frequency is so sensitive to the GCN. The slopes for the

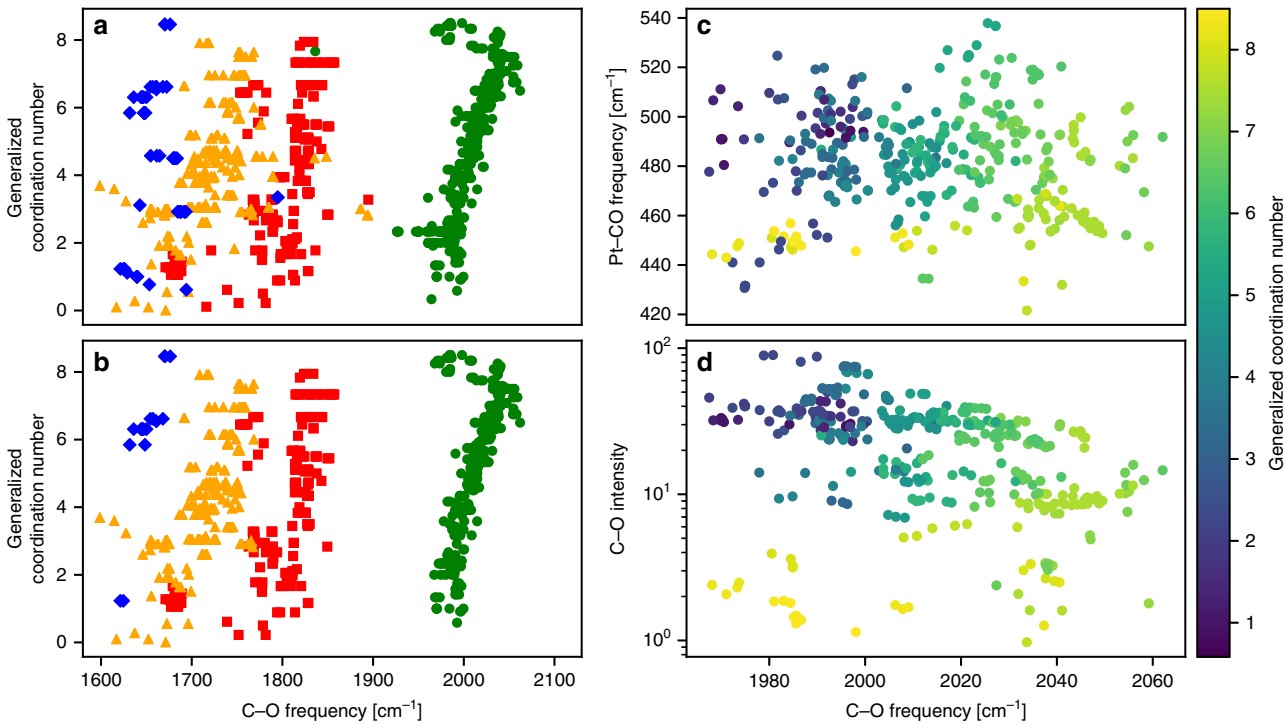

**Fig. 3 Visualization of the primary DFT dataset and removal of outliers. a–d** Data visualized in terms of the generalized coordination number (GCN), C-O frequencies, and/or intensities calculated for chemisorbed CO. **a** Primary data set. **b** Dataset after removing outliers (see text). Colors/shapes indicate GCN and frequency pairs for CO chemisorbed on atop (green/circles), bridge (red/squares), threefold (yellow/triangles) and fourfold (blue/square) sites. **c**, **d** 2D visualizations of chemisorbed Pt–CO stretch frequencies and intensities, respectively, at atop sites vs. the C–O stretch frequency. The color of the points indicates the value of the GCN depicted in the color map on the right. Colors go from low (violet) to high (yellow) values.

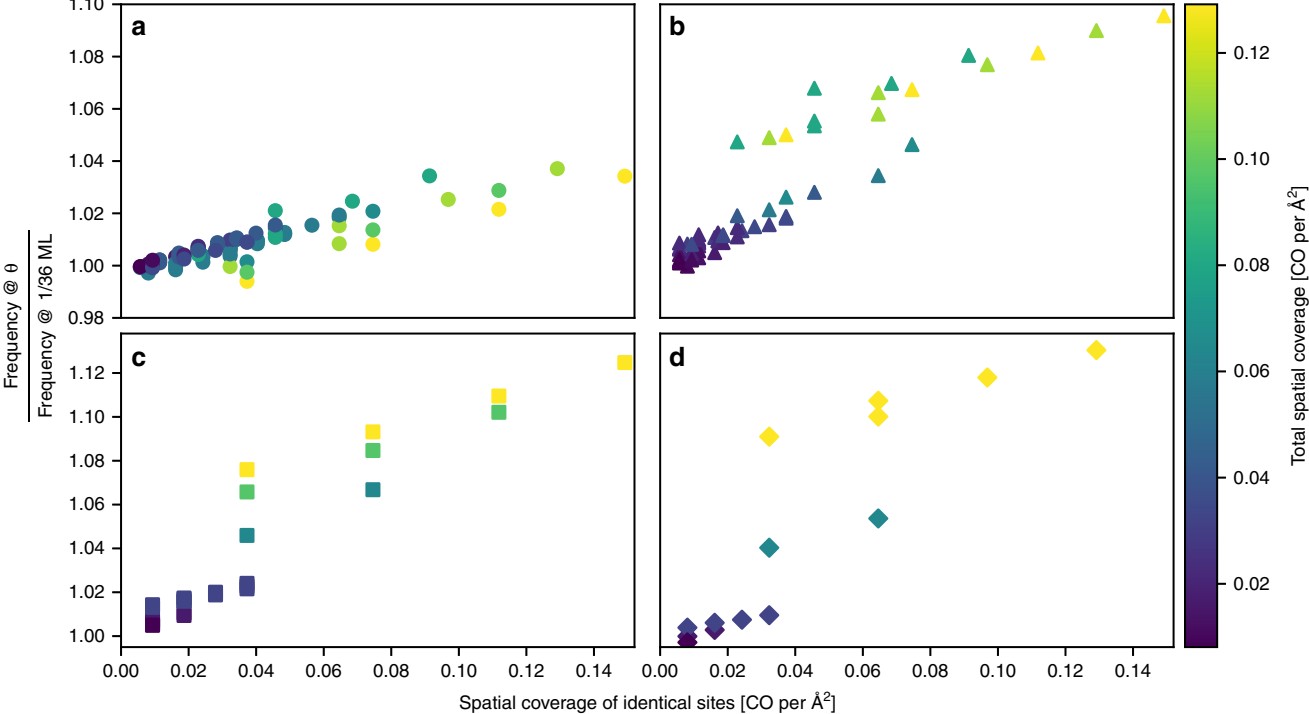

**Fig. 4 CO coverage-IR linear scaling relations.** Frequency-coverage scaling factors (y-axis) for atop (**a**), bridge (**b**), threefold (**c**) and fourfold (**d**) hollow sites. Data is regressed using ordinary least squares (OLS) on both site-specific spatial coverage (x-axis) and total spatial coverage (color variation) of CO. The $R^2$ values of the regression are 0.937 (**a**), 0.966 (**b**), 0.989 (**c**), and 0.965 (**d**). Replacing site-specific spatial coverage with site-specific relative coverage results in better statistical models for the atop and bridge bound species; however, using only spatial coverage in the model allows for enforcing spatial coverage constraints when generating the secondary dataset of synthetic spectra. Data from the 111, 100, and 110 surfaces fall on the same line for each binding-type at either constant total spatial coverage or spatial coverage of identical sites.

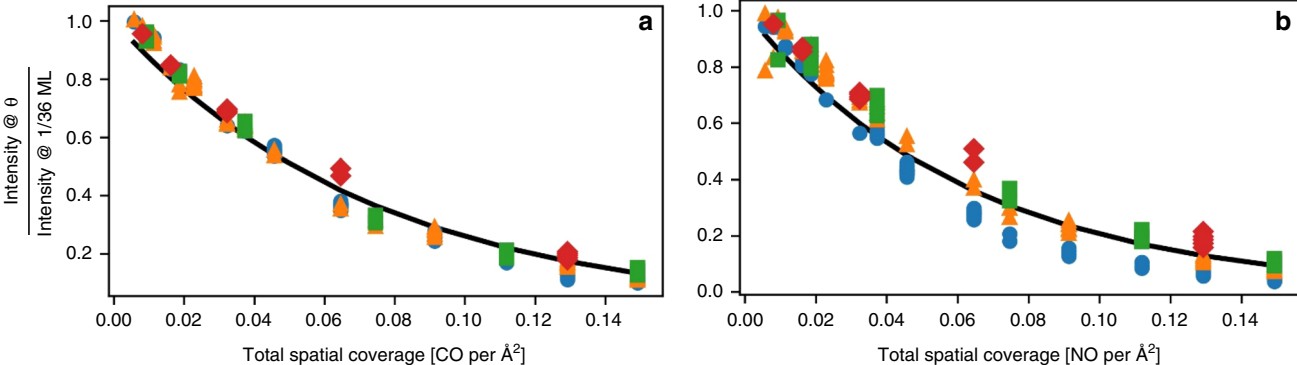

**Fig. 5 Intensity-coverage scaling factors (CSF$_I$).** Referenced to 1/36 ML coverage for the (**a**) C–O stretch and (**b**) N–O stretch modes vs. coverage. Calculations are for extended Pt surfaces with either adsorbed CO (**a**) or NO (**b**). Data for atop sites (blue circles), bridge sites (green squares), threefold sites (yellow triangles), and fourfold sites (red diamonds) all fall on the same curve. Also shown is the exponential fit of CSF$_I$ to total spatial coverage ($\theta_T$). $R^2$ values are 0.97 and 0.92 for the fits to CO and NO, respectively.

frequencies are larger for threefold and fourfold sites than for atop and bridge sites, implying frequencies for various binding-types are closer to each other at high than at low coverages.

Unlike their frequency counterpart, we find that intensity-coverage scaling factors depend only on total spatial coverage. An exponential fit of intensity-coverage scaling factor to total spatial coverage for both CO and NO on Pt is shown in Fig. 5a, b, respectively. All 173 data points in each figure fall on the same curve (including all binding-types) clearly indicating universality. Coverage scaling to the Pt–CO and Pt-NO frequencies and intensities are illustrated in Supplementary Figs. 2 and 17, respectively. Coverage scaling of N–O frequencies is almost identical to that for C–O and is given in Supplementary Fig. 16. Given that the intensity decreases with increasing coverage, it is easy to underestimate the amount of CO adsorbed on low-index planes compared to other types of sites, such as defects, when most of the surface is exposed as low-index planes. If, however, low GCN defect sites (such as adatoms) are embedded in a high coverage low-index plane, the intensity per adsorbate will also be reduced. These coverage scaling relations allow us to include lateral interactions when generating IR spectra; conversely, the shifts in IR spectra with coverage can provide an estimate of coverage on the surface in terms of both spatial uniformity and absolute coverage.

No real system of interest has a single CO molecule adsorbed. Therefore, spectral mixing is used to generate a larger dataset of synthetic spectra. Here, the primary DFT frequencies are adjusted with suitable frequency factors and the frequencies and intensities are rescaled using Figs. 4 and 5 by sampling the coverage from zero to the maximum possible spatial coverage of CO[44]. The abstraction of coverage effects we implement is to randomly apply any level of coverage shift to each set of DFT-calculated frequencies and intensities. We apply physical constraints dictating the maximum local spatial coverage of CO and ensure that the spectra selected to be mixed are consistent with the coverage effects applied. After shifting frequencies and intensities for coverage effects, the spectra are added together linearly (a process known as spectral mixing) to generate complex synthetic spectra because theory shows IR intensity is linear with non-interacting particle density[45]. After mixing, a Fourier convolution is applied to each complex spectrum in this synthetic dataset with a random line-width to account for known and unknown broadening that occur in experimental spectra. Fourier convolution of two functions involves multiplication of their Fourier transforms. The physics-driven models and sampling techniques are detailed in the Methods; Supplementary Fig. 6 contains an example of synthetic spectra. In total, hundreds of thousands of

synthetic spectra were necessary to learn the mapping between spectra and microstructure. Models trained on this data can be applied to experimental data without knowledge of the exact coverage or spatial distribution and are robust with respect to spectral broadening. The representation we use to describe microstructure and the model for learning this mapping are discussed in the next section.

**Learning structure from spectra via machine learning (ML).** There is no physical model that allows adsorption site prediction from a single set of frequencies and intensities. Learning surface microstructure is further complicated by the fact that there are many CO molecules adsorbed on an ensemble of sites. We quantify microstructure with both binding-type (a discrete distribution) and GCN (a nearly continuous distribution that we discretize) pdf. Because ML algorithms can only interpolate, the sets of pdfs and spectra we generate must adequately sample the entire population. In the Methods section, we develop random sampling of individual sites that result in distributions of binding-type and GCN pdfs that allow us to sample the whole state space as evenly as possible.

We use neural networks to solve the inverse problem of mapping IR spectra back to microstructure. The neural networks predict structure, represented by binding-type and GCN probability pdfs, using the synthetic IR spectra of the secondary dataset directly as input. Each generated complex spectrum is discretized into 500 points resulting in 4 cm$^{-1}$ intervals between 200 and 2200 cm$^{-1}$. This same discretization is ultimately applied to experimental spectra. In order to provide a measure of uncertainty in the predicted pdfs, when the model is applied to experimental spectra, ensembles of 200 neural networks are trained on synthetic spectra generated from primary DFT data. Each neural network selected for an ensemble results in roughly the same minimum cross validation error during training. These neural networks are trained using different partitions of the primary DFT data, Gaussian perturbations in the primary DFT data to account for uncertainty in the scaling factor, and different sets of hyperparameters. These ensembles therefore capture uncertainty in the model resulting from the specific primary DFT data used in generating the synthetic spectra, variance relating to the scaling factor, and the specific set of hyperparameters used. Further details of the neural network architecture and selected hyperparameters are discussed in SI Note 7.

We discuss our selection and further implementation of the neural network ensemble throughout the SI, along with the derivation of the Wasserstein loss derivative with respect to

the softmax for backpropagation of a neural network for multinomial regression in SI Note 6. We note here the importance of the Wasserstein loss; it takes into account inter-class relationships by measuring the distance between two pdfs as the minimum distance all mass of one pdf must be moved to convert it into the other. Predicting probability distributions is important in engineering and science generally, and the softmax output activation paired with the Wasserstein loss, or a loss function with similar properties, is an ideal way to do this. The closed-form solution to the derivative of the Wasserstein loss with respect to the softmax can be applied generally for learning discrete probability distributions via neural networks. Although we find that using only the high frequency portion of the spectra (intensities corresponding to frequencies above $1000 \, cm^{-1}$) reduces model accuracy (see SI Note 7), further work is needed to identify how significant each portion of the spectra is to model predictions. Interpretability of neural network decision making is itself a very active field in data science.

**Model assessment on synthetic and experimental data.** Both the binding-type and GCN pdf models can be trained with different abstractions to suit an application. Models trained strictly on synthetic spectra in the low coverage regime are the most accurate when tested on similarly generated spectra, but the least applicable to experiments performed at non-zero coverage. We design two separate neural network models for predicting the binding-type and GCN pdfs, each with their own sets of data for training and testing the models. Given the neural network models, the inverse problem is solved, where the surface microstructure, which matches the synthetic, and ultimately experimental spectra, is predicted. Details of cross validation and testing used in selecting and evaluating the best neural networks in the SI and cross validation results are given in Supplementary Fig. 8.

Occupied adsorption sites can only be identified experimentally for very simple surfaces with ordered overlayers of adsorbates on single crystals using high pressure scanning tunneling microscopy (HPSTM)[46] or low energy electron diffraction (LEED)[2] in combination with mass spectrometry (MS) and TPD. Such detailed surface science characterization is inaccessible for actual heterogeneous catalysts. Characterization of adsorbate distribution on actual heterogeneous catalysts using the trained synthetic spectra is a goal of this work. Therefore, due to lack of sufficient experimental IR spectra for which simultaneous characterization of the CO adsorption sites and coverage was done, we test the structural surrogate model with coverage effects on well-defined literature experimental HREELS and surface Raman data. The latter was scaled by frequency raised to power 2.7 to convert it to IR spectra[47] (published IR spectra did not include enough detail in the Pt–CO region to be digitized[48]). The exponent of 2.7 was determined by comparing HREEL and IR spectra for CO on Pt (111) in the $c(4 \times 2)$ configuration. We display four digitized literature experimental spectra in Fig. 6a for CO adsorbed on Pt (111) in a $c(4 \times 2)$ configuration at 0.5 ML[2] (green solid line), Pt (111) at 0.17 ML[2] (blue dotted line), on a Pt(110)-(1 × 2) surface at 1 ML[20] (yellow dashed-dotted line), and 55 nm Au/Pt core/shell nanoparticles[49] (dashed purple line). Supplementary Figure 11 enlarges the spectra where the Pt–CO and C–O stretch frequencies are located and details how the CO is arranged on the surface using LEED and other expert knowledge.

Figure 6a, b depicts the predicted binding-type and GCN pdfs, respectively, along with a 95% prediction range. Chemisorbed CO on Pt(111) is predicted to occupy 70% atop sites and 30% bridge sites. CO coverage is high on low-index planes (group 10), consistent with the experiment. This information reveals that the majority of this surface is indeed Pt(111). For the data of 0.17 ML

CO on Pt(111), all of the CO is at an atop position on a low-index plane (group 10) with a small fraction on sites with GCN values between 7.0 and 7.9, consistent with a dominant Pt(111) surface. For the data of 1 ML of CO on Pt(110), 90% of the CO is adsorbed at atop sites and 10% split evenly between bridge and hollow sites. This hollow site occupation was unexpected as it is not mentioned in literature. Closer inspection of the spectra (SI Note 8) reveals intensity contributions in the frequency ranges for hollow sites. The predicted GCN pdf is consistent with a high coverage CO on a low-index plane when uncertainty is considered. Uncertainty in the neural network ensemble prediction is bimodal, with the ensemble predicting that the CO on the (110) surface at high coverage is either at low coverage on a surface with higher coordination, such as Pt(111), or at high coverage on an extended surface of lower coordination such as Pt (110). This uncertainty could also be caused by experiments, as Pt (110) can reconstruct to an hexagonal (111)-like surface. The data for the CO on the Au/Pt core/shell nanoparticles indicates that 85% of CO is adsorbed on atop sites with 15% on bridge sites without CO being in compact ordered overlayers. Instead, most of the CO is on Pt(111) at low coverage with up to 20% adsorbed on the (100) surface. The sharp peaks in the GCN pdf are consistent with ordered surfaces. Overall, model predictions are in excellent agreement with the known microstructure and coverage of CO for the limited number of well-characterized studies and provide confidence that the proposed method can enable analysis of experimental IR data for filling in the materials gap. Underlying uncertainties in the predictions are, again, bimodal. For example, the binding-type neural network ensemble essentially predicts that CO on Pt(111) at high coverage is at some combination of only atop or bridge sites (Fig. 6b; green bars with forward slash) while the GCN group predicted by the GCN neural network ensemble predicts that 1 ML of atop-bound CO on the Pt(110) surface is adsorbed either on the (111) surface at low coverage or on high coverage on extended surfaces (Fig. 6c; yellow bars with horizontal stripes).

While the overall predictions may have bimodal uncertainty between different binding-types or GCN groups, the histogram of predictions produced by the neural network ensembles for any one group is unimodal as indicated in Fig. 7a, b. Shown are all of the individual predictions for the binding-type and GCN ensemble models with the largest predicted uncertainties.

## Discussion
The materials gap and lack of detailed surface characterization methods create one of the grand challenges in predictive science and our ability to rationally tune interfaces to improve materials' performance. While surface spectroscopic methods are sensitive, deconvolution of complex spectra is daunting due to materials heterogeneity and adsorbate coverage effects. Here we introduced a general method that can address this problem. We discover universal coverage scaling relations for converting low coverage DFT data to frequencies and intensities of systems with variable coverage that are independent of the site coordination-environment. Unexpectedly, the intensity per CO molecule decreases as the local spatial coverage increases. The interaction of the CO 1π orbital with the metal d-bands explains both the clustering of the data and the binding-type dependence for the coverage scaling relations. By leveraging the coverage-shifts of the spectra and by mixing single-adsorbate spectra, we generate hundreds of thousands of complex synthetic spectra through Fourier convolution that adequately sample the whole state space of spectra produced from CO adsorbed on Pt nanoparticles at any coverage. We demonstrate that experimental spectra of complex materials can be modeled accurately and use these synthetic

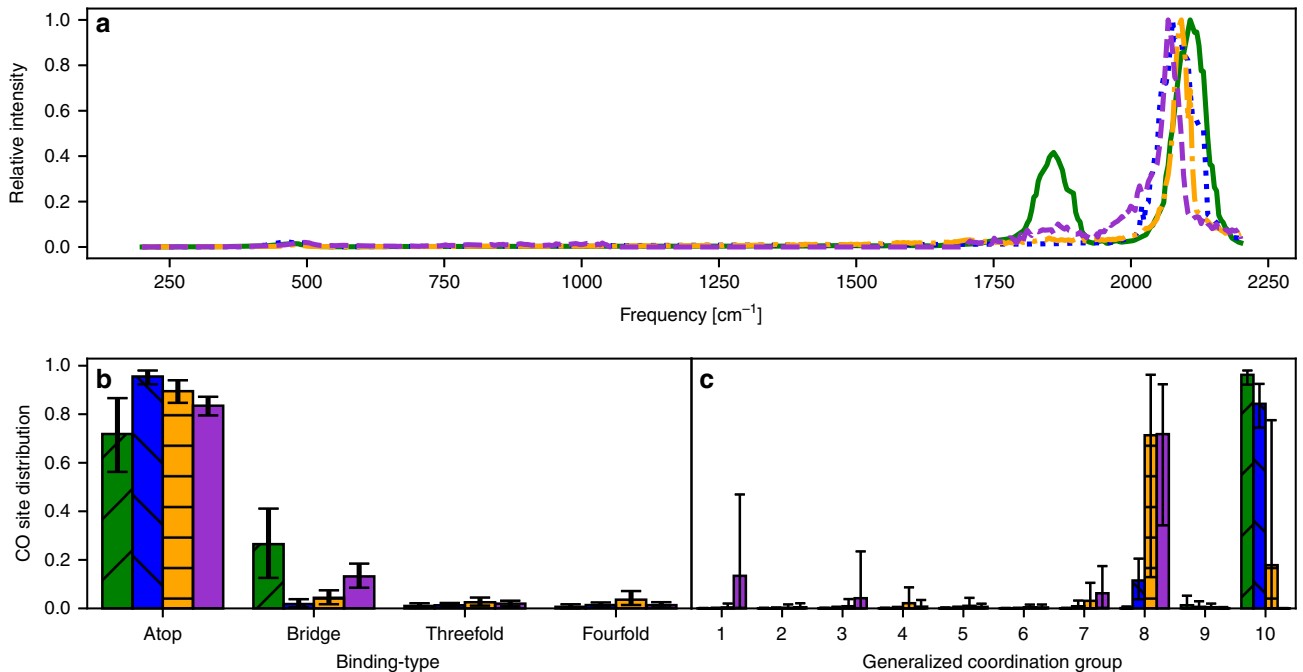

**Fig. 6 Experimental literature HREEL and SERS spectra for CO on platinum, converted to IR spectra, and predicted microstructure. a** Discretized experimental spectra. **b/c** Inverse machine learning model predictions for types and environments of sites. GCN data is binned into groups of GCN ranges: group 1 (0–1.8), 2 (1.8–2.8), 3 (2.8–3.7), 4 (3.7–4.5), 5 (4.5–5.2), 6 (5.2–6.1), 7 (6.1–7.0), 8 (7.0–7.9), 9 (7.9–8.5) and 10 (high coverage low-index planes). Spectra and predictions are for CO on Pt(111) at 0.5 ML (green solid line and bars with forward slashes (**b**, **c**)), Pt(111) at 0.17 ML (blue dotted line and bars with backslashes (**b**, **c**)), Pt(110) at 1 ML (yellow dashed-dotted line and bars with horizontal lines (**b**, **c**)), and Au/Pt core/shell nanoparticles at low coverage (purple dashed line and solid bars (**b**, **c**)). An ensemble of neural networks is used resulting in a distribution of predictions. The colored bars indicate the mean value of the prediction and the black error bars indicate the 95% prediction region.

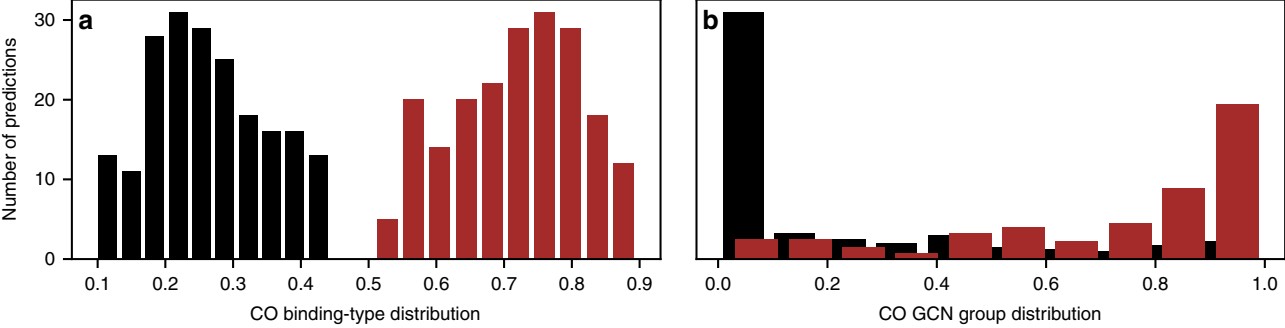

**Fig. 7 Histograms of neural network ensemble predictions with the greatest uncertainty indicated in Figure 6b, c.** Distribution of predicted binding-types for CO on Pt(111) at 0.5 ML (**a**) for atop (brown bars) and bridge (black bars). Also shown is the distribution of predicted generalized coordination number (GCN) groups (**b**) for group 8 (brown bars) and group 10 (black bars) for CO on Pt(110) at 1 ML.

spectra to train a set of neural network ensemble models to predict binding-type and GCN pdfs that represent surface coordination. We find that including the low frequency range of the spectra improves predictions and extend the method to NO on Pt nanoparticles. As most real systems exposed to adsorption do not contain a single structure, these results can be used to generate the set of most likely structures. The ability to generate these three dimensional structures will require a multimodal approach and is the subject of future work. Using first-principles based tools to aid in understanding experimental data can uncover structure and function of complex materials in order to accelerate materials design.

## Methods
**Section 1: Selecting CO as a probe molecule**. CO is extensively used as a probe molecule for IR studies because of its well-defined peaks. It is also ideal for computations because its frequencies can be computed accurately[28], its distinctive

C–O stretch frequency depends on both site type[28] and coordination[50], and its vibrational modes are driven by interaction with the metal surface[31]. We extend the model using NO as a probe molecule in the SI because NO binds in a similar manner to CO.

**Section 2: DFT calculations**. Forces used for Hamiltonians and electron densities for dipole moments were obtained using the Vienna ab initio Simulation Package (VASP) version 5.4 with the projector augmented wave method (PAWs)[51]. We selected the RPBE density functional[52] with D3 dispersion corrections[53] because it has been shown to work well for calculating properties of adsorbates[54] and accurately reproduces the Pt lattice constant (3.935 Å computed vs 3.912 Å experimental[55]). Importantly, it estimates accurate frequencies for chemisorbates[31]. For nanoparticle systems involving spin-polarized calculations, the initial magnetic moments for each atom were set to the VASP default. A Monkhorst-Pack k-point sampling grid was used for all slab calculations[56] with a single gamma point for all nanoparticle based calculations. All calculations were performed with a 400 eV plane wave cutoff and energies converged to $10^{-6}$ eV. A 0.05 eV/Å hard cut-off was applied to the carbon and oxygen atoms in their equilibrium state since all platinum atoms were frozen for the vibrational calculations.

**Section 3: Unit cell setup**. For nanoparticle calculations, the vacuum space was 20 Å between periodic images in the unit cell axes directions. For slab calculations, the vacuum space between periodic images in the surface normal direction was set to 20 Å. Image dipole corrections did not significantly alter frequency and intensity calculations for CO adsorbed on the nanoparticles but can slow convergence of the wave function so were left out. These corrections were applied to the energy, forces, and potential in the surface normal direction only for calculations involving extended surfaces. Dipole corrections were included for all calculations involving NO, including nanoparticles.

The slabs used in parameterizing the high coverage GCN pdf model were 8 atoms thick. Slabs used to identify the lateral interactions for CO were 4 atoms thick. The number atoms in each layer varied depending on the specific coverage and overlayer pattern. The number of k-points in directions perpendicular to the surface normal times the width of the unit cell in the relevant direction was at least 30 Å to maintain accuracy of the forces and dipoles. The k-point mesh was therefore set to 12/n × 12/m × 1 where n and m were the number of Pt slab atoms in the x and y direction, respectively. For all slab calculations, the top two layers were relaxed and the remaining bottom layers were fixed with a 3.935 Å lattice constant to simulate the bulk, determined using a 15 × 15 × 15 k-point grid using the tetrahedron method with Blöchl corrections combined with the Birch–Murnaghan equation of state[57]. All input files were created using the atomistic simulation environment (ASE).

**Section 4: Adsorption sites and first-principles data**. We study adsorption of CO at the three and fourfold hollow, bridge, and atop sites on nanoparticles ranging from 3 to 201 atoms. The sampled nanoparticles include icosahedrons and octahedrons as well as structures built using Wulff construction based on the experimental Pt surface energy for the (111) low-index plane[58] and the atomic corrugation factors[59] for the (100) and (110) low-index planes. Our dataset of nanoparticles also includes defected structures, fully and partially relaxed structures, as well as some systems with spin-polarized calculations. The combination of different types of structures, for which we ultimately obtain frequencies and intensities, facilitates modeling of real systems in which particle shape often diverges from the thermodynamic equilibrium. In total, we consider 1008 unique nanoparticle-adsorption systems with a single adsorbed-CO. Defected Pt nanoparticles are generated by removing Pt atoms in perfect icosahedron, octahedron, or Wulff constructed nanoparticles. Up to three atoms furthest from the chemisorbed CO were removed in order to reuse the equilibrated structure. Partial relaxation was performed by stopping a relaxation for nanoparticles with adsorbed CO before all forces on the Pt atoms are below 0.05 eV/Å.

To incorporate high coverage ordered overlayers into the GCN pdf model, first-principles data also includes the (111), (100), and (110) extended surfaces of Pt with adsorbed CO overlayers that correspond to the highest coverage pattern determined experimentally by LEED. For these facets, the highest coverage overlayers are the c(4×2)[2], c(4×2)[41], and p(2×1)[60] configurations, respectively, that correspond coverages of 0.5, 0.75, and 1 ML, as discussed extensively in literature[44].

**Section 5: Identifying the binding and describing structure**. We determine the binding according to the number of platinum atoms the carbon atom in the adsorbed carbon monoxide is connected to using a cutoff radius of 1.25 (unitless) times the van der Walls distance between atoms (Å). First, we calculate the distance between every platinum nucleus and the carbon nucleus. If the distance between the platinum and carbon atom is less than the sum of the platinum and carbon van der Walls radii (1.36 Å and 0.76 Å, respectively), we consider the two atoms to be connected. The total number of platinum atoms connected to the carbon (1, 2, 3 or 4) corresponds to classification as an atop, bridge, threefold, or fourfold site.

This same procedure (distance) used in identifying the platinum atoms connected to the carbon (binding-type) is repeated with the Pt atoms connected to the adsorption-site Pt atoms (the first nearest neighbors) to generate a site coordination. In this case, the cutoff used was 1.25 times the van der Walls diameter of a Pt atom. We use the method of Calle-Vallejo et al.[26] to calculate the GCN, which takes into account also second nearest-neighbor Pt atoms. We use the combination of binding-type and GCN to characterize adsorption microstructure. We find that the complete set of GCNs for our data is somewhat dependent on the cutoff radius with cutoff radii of 1.2, 1.25, and 1.3 all generating slightly different GCN sets. While this fact does not alter the results or their significance, alternative methods to distinguish connectivity between atoms, such as Voronoi tessellations[61] or other space filling methods, are worth exploring in future research.

**Section 6: Calculating frequencies and intensities**. Frequencies corresponding to transitions from the ground to first vibrational state were calculated using mass weighted normal mode analysis[62] under the harmonic approximation. VASP calculations provided the forces and we used these forces to construct mass weighted Hessians. Eigen decomposition was applied to the Hessians to extract eigenvalues (frequencies) and eigenvectors (directions of the vibrations). Forces on carbon and oxygen were calculated upon 0.025 Å displacements from their equilibrium positions in the x, y, and z directions. The chosen perturbation has previously been

found to be sufficient for generating spectra[63] and falls within the experimentally observed displacement due to vibration of the C–O bond in carbon monoxide's ground state[64].

IR intensities, corresponding to the normal mode frequencies, were computed by taking the matrix product of the dipole Jacobian and the normal mode eigenvectors[45]. We used the CHARGEMOL[65] charge partitioning software to integrate the electron densities provided by VASP to compute dipole moments. CHARGEMOL uses a density derived electrostatic and chemical (DDEC) approach has the ability to compute molecular polarizabilities efficiently for generating RAMAN spectra[66,67]. Separating the workflow in this way allows for improved quality control and software independence from any specific quantum mechanical code. This DFT setup used 16 surface Pt atoms and 8 CO adsorbed molecules in the periodic unit cell. When combined with scaling factors, which are discussed later, DFT does an excellent job matching experimental spectra.

**Section 7: Frequency scaling factors**. It is customary when computing frequencies to apply scaling factors to account for systematic errors between computations and experiments resulting from over (or under) binding of the functional and the assumption of a harmonic potential energy surface[29]. Unique to modeling adsorption systems is the infinite metal mass approximation. This simplification results in DFT-based frequencies associated with normal modes driven by interaction of the adsorbate with a metal surface to be systematically lower than experiment[68]. We calculate two separate scaling factors for chemisorbed carbon monoxide on platinum. The first is applied to the carbon–oxygen (C–O) stretch and the second is applied to interactions primarily driven by interaction of the metal with the adsorbate which we refer to as platinum–CO (Pt–CO) vibrations. The scaling factors ($c$) and their relative uncertainties ($u_r$) are calculated given the following equations from NIST

$$c = \frac{\sum_{i=1}^{n}(\nu_i * \omega_i)}{\sum_{i=1}^{n}\omega_i^2}, \tag{1}$$

$$u_r^2 = \frac{\sum_{i=1}^{n}\left(\omega_i^2 * \left(c - \frac{\nu_i}{\omega_i}\right)^2\right)}{\sum_{i=1}^{n}\omega_i^2}. \tag{2}$$

Equations (1) and (2) include summations over $n$ frequencies, where $\nu$ is the experimental frequency and $\omega$ is the frequency from DFT calculations. For the calculating the C–O and Pt–CO frequency scaling factors, $n$ is nine and four, respectively, and the corresponding experimental and DFT-calculated frequencies are in the Supplementary Tables 1 and 2.

**Section 8: Lateral interactions and CSFs**. We employ several methods to account for lateral interactions (coverage effects) in the binding-type and GCN pdf models. In the binding-type model, we multiply frequencies and intensities calculated from DFT at low coverage (single CO) by a CSF given both a specified coverage on a single binding-type and a total spatial coverage of all binding-types. Frequency and intensity CSFs are applied to both the C–O and Pt–CO normal modes. Both previously reported experimental studies[69] and our own computational studies show that the Pt–CO peak frequencies are not significantly affected by coverage.

There are several challenges to applying coverage effects to nanoparticle data. Packing many sites with identical GCN values on a single nanoparticle requires a very large nanoparticle and DFT calculations on large nanoparticles are very expensive. Furthermore, the reference for coverage on a nanoparticle is ill-defined. It could be referenced to all surface Pt atoms, all Pt atoms on the same facet of the nanoparticle, or all Pt atoms of the same GCN on a single facet of the Pt nanoparticle. An added benefit of using extended surfaces is that the spatial coverage of ordered overlayers is linked to the total number of Pt sites.

The frequency CSF is determined by correlating the frequency at a specific coverage divided by the frequency at 1/36 ML on a 4 × 4 Pt slab, given a specific binding-type and GCN. Correlating data from all three low-index Pt planes together, we find the coverage scaling relations to be universal, regardless of the site GCN. The CSFs generated using data from the (110) surface are nearly identical to those on the (111) surface. The intensity CSFs is computed in a similar manner. The only difference being that the intensity at 1/36 ML is multiplied by the number of CO molecules used in computing the corresponding intensity at elevated coverage. The frequency and intensity CSFs vs. coverage are illustrated in Figs. 4 and 5 of the main text and Supplementary Fig. 2.

We divide the slopes and intercepts of the regressed equations for the CSFs referenced at 1/36 ML CO on a Pt periodic slab by their respective intercepts to shift the reference to zero coverage for our nanoparticle—CO systems. The general form of the CSF and implementation is then

$$CSF_\nu = (a_{SS} * \theta_{SS} + a_T * \theta_T + 1), \tag{3}$$

$$\nu_\theta = CSF_\nu * \nu_0, \tag{4}$$

$$CSF_I = e^{a_I * \theta_T} \tag{5}$$

$$I_\theta = CSF_I * I_0. \tag{6}$$

Values of $a$ in Eqs. (3) and (5) are provided for scaling frequencies ($v$) an intensities ($I$) due to coverage ($\theta$) where coverage is given in CO atoms per $\text{Å}^2$. The frequency coverage scaling factors ($CSF_v$) are regressed using a separate OLS regression for each binding-type and regressed on site-specific spatial coverage ($\theta_{ss}$) and total spatial coverage ($\theta_T$) simultaneously. The intensity coverage scaling factor ($CSF_I$) is dependent only on $\theta_T$ with single parameter regressed via nonlinear regression. This regressed parameter ($a_I$) was found to have a value of $-13.5$ when regressed to the C–O stretch data and $-15.8$ when regressed N–O stretch data. Frequencies and intensities calculated from DFT at low coverage are given by $v_0$ and $I_0$, respectively. The scaled frequencies and intensities calculated to account for elevated coverage are given by $v_\theta$ and $I_\theta$, respectively. The regressed parameters for each $CSF_v$ are given in Supplementary Table 3. Coverage scaling is less significant for the Pt–CO frequencies with regressed parameters having values of 0.153 and $-0.258$ for $a_{ss}$ and $a_T$, respectively, resulting in an $R^2$ value of 0.520 and shown in Supplementary Fig. 3. Existing experimental studies confirm there are minimal changes to the Pt–CO frequency in response to changes in coverage. The value of $a_I$ found by applying Eq. (5) to the Pt–CO intensity data reveals a $CSF_I$ similar to that for C-O intensity scaling with a regressed value of $-13.969$ and $R^2$ of 0.938.

The frequency shifts arising from coverage effects are difficult to distinguish from differences arising from changes in GCN. For example, the un-scaled C–O DFT frequencies on Pt(111) and Pt(110) at the atop site are roughly 2046 and 2010 cm$^{-1}$ at 1/36 ML, respectively, while the shift in frequency from 0 to 1 ML at the atop site of these planes is approximately 75 cm$^{-1}$. Furthermore, while we find the normal modes of different binding-types to be largely uncorrelated, this is not the case for modes of CO at neighboring sites with slightly different GCN values if the sites are of the same binding-type. Due to the possibility of confusing coverage effects with GCN effects, CSFs developed here can only be applied to frequencies and intensities in the GCN model if the exact coverage of the spectra to be analyzed is known and there is no variation in the coverage. Furthering the difficulty of the problem, sites with different GCN values can have different coverages, even under saturation pressure. Expert knowledge of the physical system provides us with tools to address these challenges. Instead of applying CSFs in the GCN pdf model, we calculate frequencies and intensities corresponding to CO overlayers at saturation coverage because adsorption studies for the purposes of characterization are almost always performed at either very low coverage or saturation. Addressing coverage effects in this way allows us to test our model on existing experimental spectra with known occupied sites, which are exclusively at high coverage, simple surfaces with ordered overlayers due to the difficulty of isolating occupied sites for complex systems.

**Section 9: Spectral mixing**. To generate complex synthetic spectra, we mix spectra from our filtered primary dataset of nanoparticles with a single-adsorbed CO. Mixing intensities and frequencies directly is computationally difficult and inefficient. We first rebalance the primary data according to binding-types and GCN group via random selection with replacement, a common technique to rebalance imbalanced data, and then convolute using Fourier transforms all the DFT-calculated frequencies and intensities with a Gaussian filter to generate single-adsorbed CO spectra ranging from 200 to 2200 cm$^{-1}$ with a resolution of ~4 cm$^{-1}$. We can then efficiently mix spectra by applying a vector sum to the single-adsorbed CO spectra because IR spectral intensity is linear with respect to the number of molecules[45]. A Gaussian function is the initial filter because random noise results in a Gaussian signal response, although any response with a small standard deviation can be used as the goal is to map the frequencies and intensities to spectra that can be added together. To prevent significant information loss[70], the FWHM of the initial filter is set to 8 cm$^{-1}$.

It is important when generating complex spectra that we randomly sample the whole state space resulting from the mixed simple spectra, including the extremes. To generate complex spectra, we select a random number ($n$) of simple spectra to mix such that the probability of n is given by Eq. (7)

$$P(n) = \begin{cases} \frac{1}{N}; & n = 1, 2 \dots, N \\ 0; & \text{otherwise} \end{cases}. \tag{7}$$

Here $N = 200$ is sufficient as the precision of the regressed value for binding-type or GCN group by the models is larger than 0.5%. We select each simple spectra within a group with equal probability, where the probability of selecting from any particular group is also a random variable ($p_j$) whose probability is given in Eq. (8)

$$P\left(p_{j=i\neq C}\right) = \begin{cases} \frac{1}{b_i - a}; & a \leq p_i < b_i \\ 0; & p_i < a \text{ or } p_i \geq b_i \end{cases}. \tag{8}$$

In Eq. (8), $a = 0$ and

$$b_i = \begin{cases} 1; & i = 1 \\ 1 - \sum_{j<i} p_j; & i = 2, 3, \dots C-2, C-1 \end{cases}. \tag{9}$$

$C$ is the total number of groups and the probability of the group assigned with the last index is

$$p_{j=C} = 1 - \sum_{j<C} p_j. \tag{10}$$

The probability of assigning any index $j$ to any group is a discrete, uniform random variable sampled from the groups that have not already been assigned an index $j$. The generated probabilities ($p_j$), for a generic binding-type and GCN group, are given in Supplementary Fig. 4. A challenge with training the GCN model is that there are relatively few samples with large contributions to the complex spectra from any specific GCN group. This cannot be avoided as adding more groups necessarily reduces the likelihood of a significant contribution from any one group.

**Section 10: Spectral convolution**. To generate the Gaussian function for initial convolution, we take advantage of the linear relationship between FWHM and the standard deviation of a Gaussian function outlined by Valentine et al.[71] to generate the Gaussian impulse function. Every normal frequency/intensity pair is convoluted by the normalized Gaussian impulse function, integrating over the discrete range of 200–2200 cm$^{-1}$ as seen in Eq. (11),

$$\text{single spectra} = \frac{1}{\sigma\sqrt{2\pi}} \sum_{i=1}^{N} \left( I_i e^{-\frac{(v_i - E)^2}{2\sigma^2}} \right). \tag{11}$$

Here $\sigma$ is the standard deviation, $v_i$ and $I_i$ are the frequency and intensity, respectively, associated with a normal mode vibration, and $E$ is a vector of vibrational energies in wavenumbers ranging from 200 to 2200 cm$^{-1}$. $N$ corresponds to the number of normal mode vibrations resulting from each normal mode calculation.

We convolute complex spectra generated using spectral mixing with both Gaussian and Lorentzian filters, applying the method of Wertheim et al.[72] to generate the convoluting impulse. A slightly different convoluting function is applied to each combination of single spectra. The FWHM comes from a distribution given by Eq. (12)

$$P(\text{FWHM}) = \begin{cases} \frac{1}{75\,\text{cm}^{-1}}; & 2\,\text{cm}^{-1} \leq \text{FWHM} \leq 75\,\text{cm}^{-1} \\ 0; & \text{FWHM} < 2\,\text{cm}^{-1} \text{ or FWHM} > 75\,\text{cm}^{-1} \end{cases}. \tag{12}$$

The fraction of Gaussian in each convoluting function ($f_G$) is given by Eq. (13),

$$P(f_G) = \begin{cases} 1; & 0 \leq f_G \leq 1 \\ 0; & f_G < 0 \text{ or } f_G > 1 \end{cases}. \tag{13}$$

The fraction of Lorentzian in each convoluting function is then (1-$f_G$). The complex spectra is convolved with the impulse function using efficient, discrete, linear convolution.

## Data availability

Most data of this study are available within the paper and its Supplementary Information, or with the associated Python package described in the section on Code Availability. VASP and CHARGEMOL input and output files are available on Zenodo at https://zenodo.org/record/3666992#.XkYfomhKiCh as a dataset, along with trained neural networks and cross validation results. Any other data will be provided by the corresponding author upon request.

## Code availability

A software package for building the surrogate models associated with this work can be found on our group GitHub page at https://github.com/VlachosGroup/jl_spectra_2_structure. Also included in the GitHub are JSON files containing the frequencies and intensities calculated from first principles, the corresponding binding-type and GCN value of the adsorption site, and the data indices used in partitioning the stratified training and test sets for the binding model. Instructions on utilizing the package, data, and accompany scripts are included at the end of the Supplementary Information and in online documentation.

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

## Acknowledgements

This material is based upon work supported as part of the Catalysis Center for Energy Innovation, an Energy Frontier Research Center funded by the U.S. Department of Energy, Office of Science, Office of Basic Energy Sciences under Award Number DE-SC0001004. Computational time from the Blue Waters sustained-petascale computing project, which is supported by the National Science Foundation (awards OCI-0725070 and ACI-1238993) and the state of Illinois, is gratefully acknowledged. Blue Waters is a joint effort of the University of Illinois at Urbana-Champaign and its National Center for Supercomputing Applications. The 2019-2020 Blue Waters Graduate Fellowship to J.L.L. is also gratefully acknowledged.

## Author contributions

J.L.L. performed all DFT calculations, derivations and model development. D.G.V. contributed the idea of describing microstructure with IR spectra. Both authors contributed to writing.

## Competing interests

The authors declare no competing interests.
