## [Peer Review File · Nature Communications]

Reviewers' comments:

Reviewer #1 (Remarks to the Author):

Bridging the materials gap is a compelling challenge for modern catalyst design. In this work, Lansford et al. utilize analysis of complex surface micro-structures with data-assisted computational spectroscopic techniques to assess the adsorption of CO in a multitude of coordination environments including defects, ad-atoms, vacancies over Pt clusters and nano-particles. The research findings were supported by combination of physics-driven surrogate models, relevant first-principles simulations and machine learning algorithmic predictions. The problem at hand, adsorption of CO* on Pt surfaces and nano-particles, represents a bellwether system in the field, hence the study is well-aligned with the aim and scope of the journal. Utilization of a multitude of analytical techniques such as physics-based surrogate outlier screening, spectral mixing and convolution, multinomial regression via neural network for learning probability distribution functions, and quantum simulation techniques such as density functional theory is a creative combination, and is likely to appeal to the interest of the broader scientific community. However, perhaps due to the extreme breadth of techniques, the work is very difficult to read in its current form, and a substantial re-organization is recommended to more clearly present the work. Moreover, the current form of the article does not sufficiently articulate the molecular interpretations or implications of the findings, and it would be remarkably difficult for an experimentalist to apply the techniques to analyze their spectra. The current form of the article is probably better suited for a journal focusing on computational simulations such as npj Computational Materials or similar. However, with major revisions discussed below the article could be suitable for Nature Communications.

Major suggestions:

* The reliability and interpretation of the final result is not sufficiently well-articulated that an experimental spectroscopist could follow it. For example, if a researcher has an experimental IR spectra from CO* on Pt nanoparticles, what would the next steps be? and what could they expect to understand? Presumably, the method would give them an estimate of the distribution of types of binding sites and GCN's, but how reliable would it be? Is it possible to have multiple interpretations of the spectra, or is there a unique solution? More practically, how could the spectroscopist access the necessary code/data to obtain this analysis? My advice is to include a "case study" using an experimental FTIR spectra from Pt nanoparticles (possibly in collaboration with a spectroscopist), clearly describe the process of applying this technique to the independent data, and discuss of how the results should be understood by a practitioner.

* The overall structure of the manuscript is difficult to follow. Many ideas are introduced, but the key points are not clearly highlighted. For example, the authors mention several times that they have derived a closed-form solution to the derivative of the squared Wasserstein distance. While interesting, this is a relatively obscure (from the perspective of a chemist) fact that should perhaps be relegated to the supplementary information (and published separately in a mathematical journal as appropriate). It is not clear why the Wasserstein distance was chosen, or how the neural network was constructed. A schematic describing the architecture of the network (number of layers, nodes, activation functions) as well as the inputs/outputs would go a long way. Even after reading the paper several times, and looking at the SI, it is not 100% clear to me what the neural network is doing. Another particular weak point is the description of the generation of synthetic spectra. I understand that the authors used Fourier convolutions and spectral mixing to generate these "secondary" spectra, but I do not understand how they recovered a molecular interpretation of these spectra. In other words, what are the variables that are varied to generate these spectra? Presumably it is the amount/proportion of each different type of site, but it would be useful to see the specifics described in a schematic in the main text. Essentially, Fig. 1b should be expanded to include substantial details for the later steps in the workflow.

* The authors claim that their method untangles site-specific molecule/surface interactions. However, the evidence for this is not clearly presented. It seems that the method provides insight into binding site and GCN, but the ability to solve the full inverse problem (e.g. generate atomic-scale surface structures corresponding to a given spectra) is not clearly described. The method does not seem to be able to discriminate between coverage-dependent effects and coordination effects of a specific atom.

* Figure 4 exhibits CO coverage-IR linear scaling relations between frequency and spatial coverage for the atop, bridge and hollow sites. There is a drastic change in the linear scaling depending on the spatial coverage for the hollow sites. However, the scaling is described as "universal" and this drastic discontinuity is not discussed.

Minor suggestions:

* The authors have used the following sentence "This is a top priority research direction of the new catalysis research roadmap of the United States Department of Energy". The reviewer suggests replacing "top priority" with "active research direction", omitting "new" and "United States DOE" since catalysis research has broader community of readers worldwide.

* The authors claim to discover that density functional theory (DFT) generated spectra, even at high coverage, are much more accurate for determining adsorption site and deducing local microstructure than DFT energies. However, these calculations are also much more time consuming (especially for large molecules), and require some experimental comparison. It would be nice to have some discussion of the trade-off between computational time cost and accuracy, and I do not think that the strong statement on page 6, line 110 ("Clearly, spectroscopic signatures win over energy-based methods...") is fully justified. I agree that there are situations where this is the case, but there are also situations where the opposite is true.

* In page 6, line 123, "Figure 3Error! Reference source not found." needs to be fixed.

* The authors should cite the VASP software package, and include some justification of the DFT parameters chosen. The 400 eV cutoff is a bit low, layers are not relaxed, the dipole correction is not applied (?), and the RPBE-D3BJ functional is used. Are spectral features much less sensitive to these parameters than energies? Some brief discussion would be appropriate.

* On page 5, line 94 the authors state that "The intensities, corresponding to normal mode frequencies..." which sounds like the intensities are determined by the normal mode frequencies. I know what the authors meant, but found this sentence confusing.

Reviewer #2 (Remarks to the Author):

The manuscript by Dr. Vlachos et al. introduces a new methodology to predict surface sites of real nanocatalysts in terms of binding-type and generalised coordination number (GCN). The approach includes simulation of the FTIR spectra for simple systems by means of density function theory followed by application of physics-driven surrogate model to generate a secondary dataset of synthetic FTIR spectra. Two separate neural networks are trained on the synthetic dataset to calculate probability distribution functions for binding-type and GCN. Finally, the neural network could be applied to experimental IR spectra of complex material to predict microstructure operando.

The manuscript shows novel results from appealing point of view and could be of interest for a wider community of material science and catalysts. The application of physics-driven machine learning to address the surface microstructure of complex materials is of great importance for

development of new industrially-relevant catalysts. Moreover, I believe that game-changing machine learning algorithms will be standard tools in a broad range of spectroscopies in the nearest future.

The manuscript is clearly written and well-organised. The conclusions are convincing. However, I believe, there are several issues addressed could strengthen the manuscript.

1. Figure 2 shows DFT frequencies and corresponding intensities for adsorbed CO on different sites. High frequency intensities are clearly visible due to high amplitudes, while low frequency low vibration modes are less informative. Appropriate scaling of low frequency part of the figure could bring additional valuable information to the figure. Separate scale bars for high and low intensity features (as shown in Supplementary Figure 9b,c) could give better visualisation of the correlation between peaks intensities and GCN for the same binding-type.

2. Moreover, I'm curious how different are the contributions from low and high intensity peaks which are in the range of 200-700 cm^{-1} and 1600-2200 cm^{-1} , respectively. It is stated that «Each generated complex spectrum is discretised into 501 points resulting in 4 cm^{-1} intervals between 200 cm^{-1} and 2200 cm^{-1} . The same discretisation is ultimately applied to experimental spectra» (p. 10 line 189). Did I get it right that the contribution of the low intensity features (in the range of 200-700 cm^{-1}) are about two orders of magnitude lower compared to high intensity features (1600-2200 cm^{-1})? I wonder does this low intensity features make significant impact on the binding-type and, particularly, to GCN pdfs or used mainly to get scaling factors? Which vibration mode should be more sensitive to GCN from the physical point of view? Could you please comment on this issue as I feel like I didn't catch it from the manuscript and supplementary.

3. Supplementary Figure 1 shows theoretical and experimental FTIR spectra, however, there are two additional curves in the low frequency range (200-700 cm^{-1}). What are these curves? Also in the supplementary text the figure marked as Figure 7 (S p.2 line 32)

4. In the caption of the Supplementary Figure 6a it is written «Distribution of CO contribution to the spectra from each site type». However there is only one spectrum and the contributions from site types are not evident. I would expect to see several spectra.

Reviewer #3 (Remarks to the Author):

Manuscript by Lansford and Vlachos presents a new physics-driven machine learning (ML) approach to identify adsorption sites from IR spectra of adsorbate. Understanding the surface of catalysts/adsorbents at the molecular level is very important and it is one of the essential steps towards bridging the material gap.

The ML procedure itself appears to be technically sound. They used ML for correlating structural descriptors (GCN, binding type, etc.) with thousands of (hypothetical/"synthetic") IR spectra generated from DFT calculated IR frequencies obtained at low coverage on well defined sites. However, the training is based on mostly empirically corrected IR frequencies. While the presented procedure seems to be a technically reasonable tool for automatic interpretation of IR spectra, the particular application described in the manuscript (CO on Pt surfaces) may not be the best one to demonstrate the power of the method. First, DFT description of CO/Pt systems is at least questionable (see below) and, second, results are shown for somewhat trivial spectra. Therefore, I cannot recommend this paper for publication in Nat. Commun. in its present form. Authors should demonstrate the power of the method on better chosen example on or two or three examples to make it clear that it works in general and not for one particular case. Nevertheless I am sure that the manuscript in its present form would be accepted by some of the more specialized journals.

A major problem I see in the system selection. While CO/Pt is of great interest experimentally for

many reasons, it is rather complicated case for DFT to be described reliably. As authors state, DFT fails to describe adsorption energies for this system while calculated frequencies are described in a better agreement. Authors are referring to Ref. 27 for explanation at several places, stating that it can be nicely understood in terms of CO pi-orbital and metal d-bands. However, for metal-CO interaction both, binding energy and CO frequency are tightly connected – they both result from the tiny interplay between pi-donation (leading to blue-shifted CO) and pi*-backdonation (red-shifting CO). The fact that CO stretching comes out right while interaction is overestimated means that frequency appears to be in agreement with experiment due to fortuitous error cancelation.

Various scaling factors are introduced in the model. DFT calculated CO stretching frequencies are scaled on the first place and authors claim small MAE corresponding to only 0.19 and 0.33% of the experimental CO and Pt-CO values, respectively. This is truly misleading: as is apparent from Table 1 is SI, the error varies from -9 to +12 cm⁻¹ and this might be a serious problem for obtaining correct result (the problem is overlooked in the manuscript).

Method performance is demonstrated on comparison with the Raman data that are scaled to IR spectra.

Intensities are based on double-harmonic approximation which is known to be rather approximate. And they are further rescaled for increased surface coverage.

It is stated on page 6 that samples that are not local minima on PES are considered to be outliers and they are removed. I do not understand how the structure not representing minima on PES can even get into the set for calculating frequencies.

In summary, the scaling factors used might be fine to consider the effects of model errors (DFT error, harmonic approximation, higher CO coverage, etc.) but it is not clear how they affect the final results. The assumptions discussed above to build the ML model are the actual limitation here. Authors should find a better system to demonstrate that the method proposed is indeed a universal tool as is stated.

Review 1

Bridging the materials gap is a compelling challenge for modern catalyst design. In this work, Lansford et al. utilize analysis of complex surface micro-structures with data-assisted computational spectroscopic techniques to assess the adsorption of CO in a multitude of coordination environments including defects, ad-atoms, vacancies over Pt clusters and nano-particles. The research findings were supported by combination of physics-driven surrogate models, relevant first-principles simulations and machine learning algorithmic predictions. The problem at hand, adsorption of CO* on Pt surfaces and nano-particles, represents a bellwether system in the field, hence the study is well-aligned with the aim and scope of the journal.

Response:

We appreciate the reviewer's support on the system we study and relevance to the Nature Community.

Utilization of a multitude of analytical techniques such as physics-based surrogate outlier screening, spectral mixing and convolution, multinomial regression via neural network for learning probability distribution functions, and quantum simulation techniques such as density functional theory is a creative combination, and is likely to appeal to the interest of the broader scientific community.

Response:

We thank the reviewer for appreciating the complexity and broader impact of this work.

However, perhaps due to the extreme breadth of techniques, the work is very difficult to read in its current form, and a substantial re-organization is recommended to more clearly present the work. Moreover, the current form of the article does not sufficiently articulate the molecular interpretations or implications of the findings, and it would be remarkably difficult for an experimentalist to apply the techniques to analyze their spectra. The current form of the article is probably better suited for a journal focusing on computational simulations such as npj Computational Materials or similar. However, with major revisions discussed below the article could be suitable for Nature Communications.

Response:

Solving the materials gap necessitates development and application of methods from different scientific domains. After the changes to enhance usability, we believe this manuscript will appeal directly to the broad Nature Communications audience.

Action:

We have significantly reorganized the manuscript, highlighted molecular interpretations, and implemented other suggestions to make the reproducibility and application of the methods developed in this work much easier.

The reliability and interpretation of the final result is not sufficiently well-articulated that an experimental spectroscopist could follow it. For example, if a researcher has an experimental IR spectra from CO* on Pt nanoparticles, what would the next steps be? and what could they expect to understand? Presumably, the method would give them an estimate of the distribution of types of binding sites and GCN's, but how reliable would it be? Is it possible to have multiple interpretations of the spectra, or is there a unique solution? More practically, how could the spectroscopist access the necessary code/data to obtain this analysis? My advice is to include a "case study" using an

experimental FTIR spectra from Pt nanoparticles (possibly in collaboration with a spectroscopist), clearly describe the process of applying this technique to the independent data, and discuss of how the results should be understood by a practitioner.

Response:

We appreciate comments regarding usability by a spectroscopist and agree that it is crucial we facilitate usability by as wide an audience as possible. There can be multiple interpretations of the spectra as the problem of mapping spectra to structure is inherently ill-defined. In fact, even the solution of a single neural network given the same hyperparameters and primary DFT data is stochastic both due to the stochasticity in generation of the complex spectra (secondary data) and the minimization scheme for finding the best model parameters. Thus, the question of reliability is important and we summarize our action below to address these issues.

Action:

Initially, we trained a unique, but single, neural network at each coverage (low, high and variable, and a uniform monolayer) for generating binding-type and GCN probability distribution functions (pdf). To address reliability in model predictions, we replace a single neural network with an ensemble of 200 neural networks for predicting each binding-type and GCN pdf allowing us to generate prediction intervals that take into account uncertainty in the scaling factors, hyperparameters, and segmentation of the primary DFT data. We develop python scripts, as well as an entire python class, to implement the neural network ensembles we use and for generating complex spectra and making new neural network ensembles from a user's own density functional data. The entire module and accompanying scripts are uploaded to Github along with online documentation. Step by step instructions for installation and using these scripts are now provided in the Supplementary Information. We believe this will make it extremely useful to experimentalists, ensure broader impact, and provide a statistical means of evaluating the errors.

The overall structure of the manuscript is difficult to follow. Many ideas are introduced, but the key points are not clearly highlighted. For example, the authors mention several times that they have derived a closed-form solution to the derivative of the squared Wasserstein distance. While interesting, this is a relatively obscure (from the perspective of a chemist) fact that should perhaps be relegated to the supplementary information (and published separately in a mathematical journal as appropriate). It is not clear why the Wasserstein distance was chosen, or how the neural network was constructed. A schematic describing the architecture of the network (number of layers, nodes, activation functions) as well as the inputs/outputs would go a long way. Even after reading the paper several times, and looking at the SI, it is not 100% clear to me what the neural network is doing. Another particular weak point is the description of the generation of synthetic spectra. I understand that the authors used Fourier convolutions and spectral mixing to generate these "secondary" spectra, but I do not understand how they recovered a molecular interpretation of these spectra. In other words, what are the variables that are varied to generate these spectra? Presumably it is the amount/proportion of each different type of site, but it would be useful to see the specifics described in a schematic in the main text. Essentially, Fig. 1b should be expanded to include substantial details for the later steps in the workflow.

Response:

We appreciate comments regarding clarity of ideas introduced. It is important that the material in the manuscript be useful to the Nature Communications audience at large. The Wasserstein distance is

important because it accounts for inter-class relationships and is therefore crucial to learning probability distribution functions (pdfs). Predicting pdfs using data-based methods is of interest to the broader scientific community. We use the squared Wasserstein loss because its derivative with respect to the softmax function is more straightforward to derive than that of the Wasserstein itself. Although the derivation and some discussion can be found in the supplementary information, we have added clarifying text to the manuscript that can be found below.

Action:

We have significantly reorganized the manuscript. Aside from including clarifying text in the Results and Discussion, we add a section titled “Modeling Overview” where each component of the surrogate model to generate synthetic complex (secondary) spectra is mentioned and section(s) in the Methods and/or SI relevant to that component are listed. The Methods and SI sections are numbered and a table of contents is included in the SI for improved readability. We also move any existing text concerning the overall spectral model to this section. To fully explain the architecture of the neural networks used, we add several tables and text to the Supporting Information that cover what hyperparameters are explored and which ones are selected and reference these additions in the Main Text. We add the following text to explain the importance of using the Wasserstein loss.

“We note here the importance of the Wasserstein loss; it takes into account inter-class relationships by measuring the distance between two pdfs as the minimum distance all mass of one pdf must be moved to convert it into the other. Predicting probability distributions is important in engineering and science generally and the softmax output activation paired with the Wasserstein loss, or a loss function with similar properties, is an ideal way to do this. The closed form solution to the derivative of the Wasserstein loss with respect to the softmax can be applied generally for learning discrete probability distributions via neural networks.”

The authors claim that their method untangles site-specific molecule/surface interactions. However, the evidence for this is not clearly presented. It seems that the method provides insight into binding site and GCN, but the ability to solve the full inverse problem (e.g. generate atomic-scale surface structures corresponding to a given spectra) is not clearly described. The method does not seem to be able to discriminate between coverage-dependent effects and coordination effects of a specific atom.

Response:

We agree that the ultimate goal is to generate atomic-scale structures yet this cannot be done with IR alone and requires a multimodal experimental and computational approach. Although coverage dependence of frequencies adds complexity, the method can distinguish between coverage-dependent effects and coordination effects of a specific atom. This can be seen clearly by the coverage scaling data; Figure 2 of the SI shows that the Pt-C frequency changes much less with coverage than that of the C-O frequency illustrated in Figure 4 of the main text. Furthermore, both more information regarding the actual coverage of the system being explored and more accurate coverage-based surrogate models will allow greater discernment of coverage and coordination effects.

Action:

We add the following text to the Discussion Section regarding the full inverse problem.

“As most real systems exposed to adsorption do not contain a single structure, these results can be used to generate the set of most likely structures simultaneously present. The ability to generate this set of

three dimensional structures will require a multimodal approach and is the subject of ongoing and future work.”

Figure 4 exhibits CO coverage-IR linear scaling relations between frequency and spatial coverage for the atop, bridge and hollow sites. There is a drastic change in the linear scaling depending on the spatial coverage for the hollow sites. However, the scaling is described as "universal" and this drastic discontinuity is not discussed.

Response:

We appreciate identification of a confusing statement. The scaling is “universal” because it is independent of site coordination or site generalized coordination number (GCN). Furthermore, the intensity scaling relations for all binding-types fall on the same curve for both adsorbed CO and NO.

Action:

We modify the following text to clarify the extent of the universality of the frequency coverage scaling relations. “DFT calculations at different total spatial coverages on 111, 100, and 110 low-index planes of CO at the atop (circles), bridge (squares), 3-fold (triangles) and 4-fold (diamonds) sites reveal universal linear scaling of C-O and Pt-CO frequencies (**Error! Reference source not found.** and Supplementary Figure 2a) **irrespective of the coordination of the site.**” and regarding intensities “All 173 data points in each figure fall on the same curve (including all binding-types) clearly indicating universality”

The authors have used the following sentence “This is a top priority research direction of the new catalysis research roadmap of the United States Department of Energy”. The reviewer suggests replacing "top priority" with "active research direction", omitting "new" and "United States DOE" since catalysis research has broader community of readers worldwide.

Response:

We appreciate the help to get across the broad benefit of our work.

Action:

We make these changes in the text.

The authors claim to discover that density functional theory (DFT) generated spectra, even at high coverage, are much more accurate for determining adsorption site and deducing local microstructure than DFT energies. However, these calculations are also much more time consuming (especially for large molecules), and require some experimental comparison. It would be nice to have some discussion of the trade-off between computational time cost and accuracy, and I do not think that the strong statement on page 6, line 110 ("Clearly, spectroscopic signatures win over energy-based methods...") is fully justified. I agree that there are situations where this is the case, but there are also situations where the opposite is true.

Response:

We acknowledge there can be situations where energy based methods can be better for determining adsorption sites when there are well defined adsorption sites. If the available adsorption sites are not known *a priori*, it is impossible to use energy-based methods without some type of molecular dynamics or Monte Carlo methods which have their own set of challenges. Besides being computationally expensive, these methods are designed for systems that evolve naturally and are therefore difficult to use when an experimentalist is designing specific kinds of sites or nanoparticles.

Action:

We have modified the cited text and added further clarification in the main text (given below).

In both accuracy and resolution, spectroscopic signatures (DFT and/or experimental IR) win over energy-based methods (DFT energies, chemisorption, TPD, and calorimetry) for closing the materials gap. A challenge DFT-based vibrational calculations face is that computational time becomes prohibitively high for very large molecules.

In page 6, line 123, "Figure 3Error! Reference source not found." needs to be fixed.

Response:

Thanks for pointing out this error.

Action:

We have fixed this and ensured that there are not any similar errors.

The authors should cite the VASP software package, and include some justification of the DFT parameters chosen. The 400 eV cutoff is a bit low, layers are not relaxed, the dipole correction is not applied (?), and the RPBE-D3BJ functional is used. Are spectral features much less sensitive to these parameters than energies? Some brief discussion would be appropriate.

Response:

We cite VASP in this line "Forces used for Hamiltonians and electron densities for dipole moments were obtained using the Vienna ab initio Simulation Package (VASP) version 5.4 with the projector augmented wave method (PAWs).¹⁷" We find that the 400 eV cutoff is sufficient and is a common cutoff in literature, and only the top two layers of extended surface are relaxed, and dipole corrections are applied to all extended surface calculations. On extended surfaces, frequencies are not sensitive to dipole induced by periodic images but intensities are very sensitive to these. We have discussed this in section 3 of the Methods and illustrate the impact of spin and dipole corrections to frequencies and intensities of adsorbates on nanoparticles in SI section 9 but add further clarification with text below.

Action:

The following text is provided or added in the Methods section.

"Image dipole corrections did not significantly alter frequency and intensity calculations for CO adsorbed on the nanoparticles but can slow convergence of the wave function so were left out. These corrections were applied to the energy, forces, and potential in the surface normal direction only for calculations involving extended surfaces. Dipole corrections were included for all calculations involving NO, including nanoparticles. Regarding the relaxation see the following existing and added text in Methods Section 3 "For all slab calculations, the top two layers were relaxed and the remaining bottom layers were fixed with a 3.935 Å lattice constant to simulate the bulk"

On page 5, line 94 the authors state that "The intensities, corresponding to normal mode frequencies..." which sounds like the intensities are determined by the normal mode frequencies. I know what the authors meant, but found this sentence confusing.

Response:

Thank you for pointing this out.

Action:

We have removed the text “, corresponding to normal mode frequencies” as it was superfluous.

Review 2

The manuscript shows novel results from appealing point of view and could be of interest for a wider community of material science and catalysts. The application of physics-driven machine learning to address the surface microstructure of complex materials is of great importance for development of new industrially-relevant catalysts. Moreover, I believe that game-changing machine learning algorithms will be standard tools in a broad range of spectroscopies in the nearest future. The manuscript is clearly written and well-organised. The conclusions are convincing. However, I believe, there are several issues addressed could strengthen the manuscript.

Response

Thank you for your support and we are glad you found our manuscript interesting. Your suggestions have made our work better.

Figure 2 shows DFT frequencies and corresponding intensities for adsorbed CO on different sites. High frequency intensities are clearly visible due to high amplitudes, while low frequency low vibration modes are less informative. Appropriate scaling of low frequency part of the figure could bring additional valuable information to the figure. Separate scale bars for high and low intensity features (as shown in Supplementary Figure 9b,c) could give better visualisation of the correlation between peaks intensities and GCN for the same binding-type.

Response

Thank you for this suggestion.

Action:

We have added a zoomed in section for the low frequencies with its own scale bars in Figure 2 in a similar manner as Figure 9b,c.

Moreover, I'm curious how different are the contributions from low and high intensity peaks which are in the range of 200-700 cm^{-1} and 1600-2200 cm^{-1} , respectively. It is stated that ² (p. 10 line 189). Did I get it right that the contribution of the low intensity features (in the range of 200-700 cm^{-1}) are about two orders of magnitude lower compared to high intensity features (1600-2200 cm^{-1})? I wonder does this low intensity features make significant impact on the binding-type and, particularly, to GCN pdfs or used mainly to get scaling factors? Which vibration mode should be more sensitive to GCN from the physical point of view? Could you please comment on this issue as I feel like I didn't catch it from the manuscript and supplementary.

Response

You are right that the intensity of the low frequency modes can be two orders of magnitude less than the intensities of the high frequency modes, although this alone does not signify that low frequency intensities are not utilized by the neural networks to predict binding-type and coordination. We find that, although the Pt-CO frequency is sensitive to GCN, the C-O frequency is sensitive in a more systematic way. We believe this has to do with interaction of the metal sp states to the adsorbate sigma

orbitals. Understanding the origin of the frequency dependence on GCN is the subject of ongoing work. For clarification and to further understand the effect of the low frequency intensities we have done the following.

Action:

We have added the following text to the discussion of the machine learning model for structure prediction. “Although we find that using only the high frequency portion of the spectra (intensities corresponding to frequencies above 1000 cm^{-1}) reduces model accuracy (see SI Section 7), further work is needed to identify how significant each portion of the spectra is to model predictions. Interpretability of neural networks is itself a very active field in AI.” In SI Section 7, we add Supplementary Figures 9 and 10 which illustrate that the loss increases when the low frequency portion of the spectra is excluded. We have also modified the text in the discussion section to include the statement “We find that including the low frequency range of the spectra improves predictions and extend the method to NO on Pt nanoparticles.”

Supplementary Figure 1 shows theoretical and experimental FTIR spectra, however, there are two additional curves in the low frequency range (200-700 cm^{-1}). What are these curves? Also in the supplementary text the figure marked as Figure 7 (S p.2 line 32)

Response

The portion of the spectra in the low frequency range corresponds to the Pt-CO vibrational modes. The higher portion of the low frequencies corresponds to atop CO and the lower portion of the frequencies corresponds to bridge CO.

Action:

We have modified the text above the figure such that it reads “The computed spectra is accurate in both the high-frequency C-O modes and the lower frequency Pt-CO modes for both atop and bridge sites.” We have corrected p.2 line 32 to read Figure 1 instead of Figure 7.

In the caption of the Supplementary Figure 6a it is written “Distribution of CO contribution to the spectra from each site type”. However there is only one spectrum and the contributions from site types are not evident. I would expect to see several spectra.

Response

The spectra in Figure 6a is the summation of the individual spectra which is why there is only one spectrum. Although, as you state, contributions are not evident by the human eye, the neural network prediction is accurate.

Action:

We have modified the captions for clarity and added the following text above the figure “In **Error! Reference source not found.a**, the peaks around 2100 cm^{-1} , 1850 cm^{-1} , and 1700 cm^{-1} correspond to contributions from occupied atop, bridge, and hollow sites, respectively. The exact contributions of these sites as well as the contributions predicted by the model are given in **Error! Reference source not found.b** by the green bars with slanted lines and the purple bars with horizontal lines, respectively.”

Review 3

Manuscript by Lansford and Vlachos presents a new physics-driven machine learning (ML) approach to identify adsorption sites from IR spectra of adsorbate. Understanding the surface of catalysts/adsorbents at the molecular level is very important and it is one of the essential steps towards bridging the material gap.

The ML procedure itself appears to be technically sound. They used ML for correlating structural descriptors (GCN, binding type, etc.) with thousands of (hypothetical/"synthetic") IR spectra generated from DFT calculated IR frequencies obtained at low coverage on well defined sites.

Response:

We agree that understanding the interaction of the surface with adsorbates is crucial to solving the materials gap.

However, the training is based on mostly empirically corrected IR frequencies. ... second, results are shown for somewhat trivial spectra...Method performance is demonstrated on comparison with the Raman data that are scaled to IR spectra.

Response:

We agree that it would be ideal to test the model on experimental IR spectra of CO on nanoparticles. Several issues arise. First, there are no current methods to determine the proportion of occupied binding sites on nanoparticles to which we can test our method. Second, as we are pulling spectra from literature and digitizing spectra, the detail on the published IR spectra in the low frequency range is not conducive to digitization. For this reason we use empirically corrected HREEL spectra of CO on extended surfaces (where coordination of adsorption can be determined from LEED, TPD, and expert knowledge) and surface enhanced Raman of CO on Pt nanoparticles (where coordination of adsorption sites can be guessed but is not known). We therefore demonstrate two of the models (high coverage binding-type and GCN models) on four different experimental spectra. Supplementary Figures 7, 8, 17, 18, 20, and 21 demonstrate the predictive power of the structure surrogate models on non-trivial synthetic spectra. The challenge with non-trivial experimental spectra is that there are no existing tools to determine what the adsorption sites are in the presence of adsorbates.

While the presented procedure seems to be a technically reasonable tool for automatic interpretation of IR spectra, the particular application described in the manuscript (CO on Pt surfaces) may not be the best one to demonstrate the power of the method. First, DFT description of CO/Pt systems is at least questionable (see below)... Therefore, I cannot recommend this paper for publication in Nat. Commun. in its present form. Authors should demonstrate the power of the method on better chosen example on or two or three examples to make it clear that it works in general and not for one particular case. Nevertheless I am sure that the manuscript in its present form would be accepted by some of the more specialized journals... Authors should find a better system to demonstrate that the method proposed is indeed a universal tool as is stated.

Response:

We appreciate your advice on adding evidence of generalizability. CO is often used as a probe molecule in experimental literature because it has well-defined peaks that are easily interpreted by people. Machine learning methods do not have the same limitations and can use more complex spectra for

understanding surface structure. We believe that the following changes to the manuscript will broaden its appeal even further and should be well-received by the Nature Communications audience.

Action:

To add evidence of generalizability, we extend the methods developed here for CO also to NO as a probe molecule in Section 10 of the Supplementary Information.

A major problem I see in the system selection. While CO/Pt is of great interest experimentally for many reasons, it is rather complicated case for DFT to be described reliably. As authors state, DFT fails to describe adsorption energies for this system while calculated frequencies are described in a better agreement. Authors are referring to Ref. 27 for explanation at several places, stating that it can be nicely understood in terms of CO pi-orbital and metal d-bands. However, for metal-CO interaction both, binding energy and CO frequency are tightly connected – they both result from the tiny interplay between pi-donation (leading to blue-shifted CO) and pi*-backdonation (red-shifting CO). The fact that CO stretching comes out right while interaction is overestimated means that frequency appears to be in agreement with experiment due to fortuitous error cancelation.

Response:

It is well known that DFT with GGA functionals does not calculate binding energies of CO on Pt extended surfaces correctly, even predicting that adsorption at the three-fold site has lower energy than the atop site. While it is true that adsorption energy is dominated by pi*-backdonation, as cited in the manuscript, work of Dabo et al.² shows that the difference in C-O frequency at different binding-types (atop, bridge, etc.) is primarily due to interaction of the CO 1 π states with the metal d-band and therefore and not back-donation. For example, even though bridge sites bind CO more weakly than atop sites, the CO frequency is lower on bridge sites than on atop sites. A plot of C-O frequency with CO binding energy reveals no correlation. We believe that interaction of the CO σ states with the metal sp-band drives changes in frequency across the same binding-types with different coordination environments and is the subject of ongoing work. Thus CO frequencies are accurate on surfaces because the governing physics is different from that for adsorption energies.

Action:

As discussed previously, we find evidence of generalizability and extend the method to NO as a probe molecule in Section 10 of the Supplementary Information. We add the following emphasis to the main text “CO frequencies do not correlate with adsorption energy as the governing physics is different (see Supplementary Figure 14).” Supplementary Figure 14 shows that CO frequencies and adsorption energies are not correlated on nanoparticles.

Various scaling factors are introduced in the model. DFT calculated CO stretching frequencies are scaled on the first place and authors claim small MAE corresponding to only 0.19 and 0.33% of the experimental CO and Pt-CO values, respectively. This is truly misleading: as is apparent from Table 1 is SI, the error varies from -9 to +12 cm⁻¹ and this might be a serious problem for obtaining correct result (the problem is overlooked in the manuscript)... Intensities are based on double-harmonic approximation which is known to be rather approximate. And they are further rescaled for increased surface coverage. It is stated on page 6 that samples that are not local minima on PES are considered to be outliers and they are removed. I do not understand how the structure not representing minima on PES can even get into the set for calculating frequencies....In summary, the scaling factors used might be fine to consider the effects of model errors (DFT error, harmonic approximation, higher CO coverage,

etc.) but it is not clear how they affect the final results. The assumptions discussed above to build the ML model are the actual limitation here.

Response:

It was not our intention to be misleading with representing error as a percentage, although we see how it could be. The intention was to show that the relative error of the Pt-CO and C-O frequencies are similar. It is common practice to rescale intensities to account for errors in the harmonic approximation and NIST has an entire database of scaling factors for gas-phase frequencies. Due to metal nanoparticles/surfaces requiring use of plane waves for computational efficiency, an analytical gradient is not possible. Therefore, small errors can result in adsorption sites with maximum forces below 0.05 eV/Å to have imaginary frequencies and therefore not be local minima on the PES. These are the majority of the outliers. We appreciate the comment on model errors and believe that addressing these comments has greatly improved our manuscript.

Action:

To remove any notion that the errors may be misleading, we add the following text to the manuscript “Although the average error is small, the error in the C-O frequency ranges from -8.92 cm^{-1} to 11.87 cm^{-1} . However, it is possible that experimental error (including uncertainty in the actual coverage) could be a contributing factor to data points with large error.” To address uncertainty in the predictions, we treat the scaling factors as random variables that follow a Gaussian distribution. We sample different scaling factors from these distributions for each frequency and corresponding intensity when generating the complex synthetic spectra. As we increase the primary data using resampling and run up to 300 sets of training data, we end up sampling this distribution many times. Ultimately, we use an ensemble of 200 neural networks for predicting each binding-type and GCN distribution allowing us to generate prediction intervals that take into account uncertainty in the scaling factors, hyperparameters, and segmentation of the primary DFT data.

- 1 Kresse, G. & Furthmüller, J. Efficient iterative schemes for ab initio total-energy calculations using a plane-wave basis set. *Phys. Rev. B*. **54**, 11169-11186 (1996).
- 2 Dabo, I., Wieckowski, A. & Marzari, N. Vibrational Recognition of Adsorption Sites for CO on Platinum and Platinum–Ruthenium Surfaces. *J. Am. Chem. Soc.* **129**, 11045-11052, doi:10.1021/ja067944u (2007).

REVIEWERS' COMMENTS:

Reviewer #1 (Remarks to the Author):

The authors have clearly taken all comments into consideration and improved the presentation and usability of the manuscript. I have a few minor comments, but overall feel that the revised version of the manuscript is comprehensive, highly impactful, and suitable for publication in Nature Communications.

* On page 2, the authors state "This is an active research direction of the United States Department of Energy." I think this is a bit narrow, since other countries and funding agencies are also interested in this issue.

* On page 6 line 122 the authors state that the "...frequency ranges from -8.92 cm^{-1} ". I think this is a typo, since the frequency should not be negative.

* I find the sentence on page 13 L266-267 a bit confusing. The authors state that the ensemble predicts that the CO is either "low coverage on a surface with higher coordination" or "high coverage on an extended surface". However, I would generally expect that an extended surface would have high coordination. Perhaps there is an error? If not, I think more explanation or different phrasing is needed.

Reviewer #2 (Remarks to the Author):

The manuscript shows novel results from appealing point of view and could be of interest for a wider community of material science and catalysis. The application of physics-driven machine learning to address the surface microstructure of complex materials is of great importance for development of new industrially-relevant catalysts. Moreover, I believe that game-changing machine learning algorithms will be standard tools in a broad range of spectroscopies in the nearest future. The manuscript is clearly written and well-organised. The conclusions are convincing.

I believe, that the manuscript after applied modifications is suitable for publication in nature communication journal.

Reviewer #3 (Remarks to the Author):

I appreciate the effort and changes authors have undertaken to improve the manuscript based on suggestions of reviewers. In particular, authors have now demonstrated proposed protocol on other than CO probe molecule (NO data added to SI). And this was my major objection. They have also carefully explained all criticized inconsistencies/unclear formulations. I am please that I can recommend the paper for publication in its present form.

Review 1

On page 2, the authors state "This is an active research direction of the United States Department of Energy." I think this is a bit narrow, since other countries and funding agencies are also interested in this issue.

Response:

We agree that other countries and funding agencies are interested in this topic and have changed the text to read "This is an active research direction of both government funding agencies and private companies"

On page 6 line 122 the authors state that the "...frequency ranges from -8.92 cm^{-1} ...". I think this is a typo, since the frequency should not be negative.

Response:

Our text states "the error in the C-O frequency ranges from -8.92 cm^{-1} to 11.87 cm^{-1} ". To avoid confusion we have changed the text to read "individual C-O frequency errors range from -8.92 cm^{-1} to 11.87 cm^{-1} "

I find the sentence on page 13 L266-267 a bit confusing. The authors state that the ensemble predicts that the CO is either "low coverage on a surface with higher coordination" or "high coverage on an extended surface". However, I would generally expect that an extended surface would have high coordination. Perhaps there is an error? If not, I think more explanation or different phrasing is needed.

Response:

We have rephrased the text to read CO "is either at low coverage on a surface with higher coordination, such as Pt(111), or at high coverage on an extended surface of lower coordination such as Pt(110)"